

# Technical note: A procedure to clean, decompose and aggregate time series

François Ritter[1]

[1]Finres, 59 Boulevard Exelmans, 75016 Paris, France

*Correspondence to*: François Ritter (ritter.francois@gmail.com)

**Abstract.** Errors, gaps and outliers complicate and sometimes invalidate the analysis of time series. While most fields have developed their own strategy to clean the raw data, no generic procedure has been promoted to standardize the pre-processing. This lack of harmonization makes the inter-comparison of studies difficult, and leads to screening methods that are usually ambiguous or case-specific. This study provides a generic pre-processing procedure (called *past*, implemented in R) dedicated

to any univariate time series. *Past* is based on data binning and decomposes the time series into a long-term trend and a cyclic component (quantified by a new metric, the Stacked Cycles Index) to finally aggregate the data. Outliers are flagged with an enhanced Boxplot rule called *Logbox*. Three different Earth Science datasets (contaminated with gaps and outliers) are successfully cleaned and aggregated with past. This illustrates the robustness of this procedure that can be valuable to any discipline.

## 1 Introduction

In any discipline, raw data need to be inspected and evaluated during a pre-processing procedure before performing the analysis. Errors are removed, values that deviate from the rest of the population are flagged (outliers, see Aguinis et al., 2013), in some cases gaps are filled. Because the raw data are altered, pre-processing is a delicate and time-consuming task that is commonly neglected due to cognitive biases deflecting our understanding of reality ('I see what I want to see'). The fate of

extreme values is crucial as they usually challenge scientific or economic theories (Reiss et al., 1997).

Time series are particularly difficult to pre-process (Chandola et al., 2009). A value can or cannot be considered as an outlier just depending on its timestamp (e.g., a cold temperature in summer); large data gaps are common (e.g., instrument failure), abrupt changes can occur (e.g., stock market crash) and a background noise frequently covers the true signal. This complexity explains why there currently is no consensus on which procedure to use even in the simple univariate case. A recent review

(Ranjan et al., 2021) covered more than 37 preprocessing methods for univariate time series, and Aguinis et al. (2013) listed 14 different outlier definitions that are mutually exclusive. Despite this (overwhelming) abundance of methods, there are

paradoxically very few R packages that offer a pre-processing function. It is worth mentioning *tsclean* or *tsoutliers* (package *forecast*, Hyndman et al., 2020) that applies a Seasonal and Trend decomposition using Loess (STL, Cleveland et al., 1990) to data showing a seasonal pattern. Outliers present in in the residuals are then flagged using the Boxplot rule (Tukey, 1977).

Another alternative is *outlierMAD* (package *pracma*, Borchers, 2021) that applies a Hampel filter (Pearson, 2002) to the time series and flag outliers based on the Mean Absolute Deviation (MAD). These functions are case-specific and will occasionally produce errors of type I (real extreme events are cut) or type II (real outliers are missed) depending on the noise level and data structure.

This study offers a more generic pre-processing procedure (called *past*, implemented in R) dedicated to univariate time series

that are particularly messy (with outliers, data gaps, missing values or irregular timesteps). In opposition to methods that apply a model to forecast future events, past makes no assumption on the structure of the signal and is only based on data binning. The time series is divided into a sequence of non-overlapping time intervals of equal period (called bins) that fulfil four purposes:

i)      Data cleaning: bins with insufficient data are discarded, and outliers are flagged in the remaining bins. If there is a

cyclic pattern within each bin, missing values can be imputed as well.

ii)     Decomposition: the timeseries is decomposed into a long-term trend and a cyclic component.

iii)    Cyclicity analysis: the mean cycle of the stacked bins is calculated, and the strength of the cyclicity is quantified by a novel index, the Stacked Cycles Index.

iv)     Aggregation: data are averaged (or summed) within each bin.

The inputs of the past procedure offer a large flexibility in terms of imputation level or outlier cutoff, but also in the timestamp of the bins: a day does not necessarily start at midnight or a year the 1st of January. The timeline is not limited to daily or monthly data but can vary from milliseconds to millenaries. The outputs keep track of the changes brought to the data: contaminated bins are flagged, as well as outliers and imputed data points. More importantly, extensive effort has been made toward making the procedure feel intuitive to an unexperienced user who should be able to understand the algorithm from a

few examples.

This paper is divided into two distinct parts. The first part improves the boxplot rule that flags outliers in univariate datasets. This enhanced version (called *LogBox*) takes the sample size into account, and its performance is compared to four other methods in the literature, including the MAD. The second part describes the past procedure, and then applies it to three datasets





that have been contaminated beforehand to show the efficiency of the algorithm. Limitations are discussed, and finally good

practice recommendations are brought in the conclusion.

## 2 Part I, outliers

### 2.1 Context

This first part is dedicated to the detection of outliers present in univariate datasets (without the time component). The boxplot (or Tukey's) rule is a commonly used method to flag outliers below a lower boundary $l$ and above an upper boundary $u$ (Tukey,

60   1977):

$$\begin{cases} l = q(0.25) - \alpha \times \big(q(0.75) - q(0.25)\big) \\ u = q(0.75) + \alpha \times \big(q(0.75) - q(0.25)\big) \end{cases}$$

With $q$ the sample quantile (e.g., $q(0.5)$ is the median) and $\alpha = 1.5$ a constant that corresponds to 99.3% of Gaussian data falling within $[l, u]$. This method is simple and robust to the presence of a maximum of 25% of outliers in the dataset (known as the breakdown point). However, two issues emerge from this rule:

(i)    For a Gaussian population, $\alpha = 1.5$ is inappropriate for large sample sizes ($n \geq 10^3$), because the number of points erroneously flagged as outliers increases linearly with $n$ (due to the 99.3% of data captured by $[l, u]$).

(ii)   For a non-Gaussian population, $\alpha = 1.5$ is generally too restrictive. For example, $\sim 4.8$ % of data following an Exponential distribution would be erroneously flagged as outliers.

Two studies have attempted to address the second issue (Kimber, 1990; Hubert & Vandervieren, 2008) by adjusting $\alpha$ to the

skewness $S$ (third standardized moment related to the asymmetry of a distribution) while ignoring the excess kurtosis $\kappa_{ex}$ (fourth standardized moment related to the tail weight). Other studies have corrected biases emerging at small sample sizes (Carling, 2000; Schwertman et al., 2004); however, none have designed a method based on the boxplot rule that can handle outliers present in large sample sizes.

A more generic method (called *LogBox*) has been developed in this study to assign $\alpha = k \log(n) + 1$ with $n$ the sample size

and $k$ a positive number that corresponds to the nature of the distribution (e.g., $k = 0.16$ for Gaussian data; $k = 0.8$ for Exponential data). A default value of $k = 0.6$ has been determined with an ensemble of non-Gaussian distributions (the Pearson family) that represent univariate datasets with moderate $S$ and $\kappa_{ex}$ (Fig. 1). A comparison with four other existing models is then performed to test the resistance of each method to different types of distributions and different sample sizes (Fig. 2).

Finally, *LogBox* is implemented (with the value of $k$ left to the user) in the aggregation procedure described in part II to clean the residuals obtained after fitting the univariate timeseries with a robust and nonparametric method.





## 2.2 Method

### 2.2.1 Definition of $\alpha^-$ and $\alpha^+$ with the 3σ, 4σ and 5σ convention

Let $D_X$ be a probability distribution of a single random variable $X$ associated with the population quantile function $Q$. Two

strictly positive functions $\alpha^-$ and $\alpha^+$ attached to $D_X$ are defined for $0.75 < p < 1$:

$$\alpha^-(p) = \frac{Q(0.25) - Q(1-p)}{Q(0.75) - Q(0.25)}$$

$$\alpha^+(p) = \frac{Q(p) - Q(0.75)}{Q(0.75) - Q(0.25)}$$

The boxplot rule can now be expressed with $\alpha^-$ and $\alpha^+$:

$$\begin{cases} l = q(0.25) - \alpha^-(p_{lim}) \times \big(q(0.75) - q(0.25)\big) \\ u = q(0.75) + \alpha^+(p_{lim}) \times \big(q(0.75) - q(0.25)\big) \end{cases}$$

With $q$ the sample quantile and $p_{lim}$ related to the percentage of data falling within $[Q(1 - p_{lim}), Q(p_{lim})]$, independent from

the nature of the distribution.

In order to set a framework consistent with the Gaussian case, we derive three $p_{lim}$ values ($p_{3\sigma}, p_{4\sigma}$ and $p_{5\sigma}$) expressed as $p_{j\sigma}$

= $\Phi(j)$ with $\Phi$ the cumulative distribution function of the standard Normal distribution $\mathcal{N}(0,1)$ and $j = \{3,4,5\}$ implicit

throughout the study. These $p_{lim}$ values are associated with the percentage of Gaussian data captured by $\pm j\sigma$ (known as the

"sigma-Rule"), with $\sigma$ the standard deviation of the Gaussian. The corresponding $\alpha_{j\sigma}^{\mathcal{N}} = \alpha^+(p_{j\sigma}) = \alpha^-(p_{j\sigma})$ values are

computed in the Gaussian case with $Q = \Phi^{-1}$ (Table 1).

Due to their structure based on a ratio of differences of quantiles (Brys et al., 2006), the functions $\alpha^-$ and $\alpha^+$ are *location* and

*scale* invariant: $D_X$ and $D_{\lambda X + \upsilon}$ share same $\alpha^-$ and $\alpha^+$ (with $\upsilon \in \mathbb{R}$ and $\lambda > 0$). Moreover, $\alpha^-$ and $\alpha^+$ are *swapped* for $\lambda < 0$:

the pair ($\alpha^-$, $\alpha^+$) for $D_X$ becomes ($\alpha^+$, $\alpha^-$) for $D_{\lambda X + \upsilon}$. These properties lead to two simplifications in this study:

1)  The *location* and *scale* parameters of a distribution have no effect on $\alpha^-$ and $\alpha^+$ therefore only the *shape* parameter

will vary, affecting the skewness $S$ and excess kurtosis $\kappa_{ex}$.

2)  Only symmetrical or right-skewed distributions will be considered ($S \geq 0$). If $D_X$ was left-skewed, then $D_{-X}$ would

be right-skewed and ($\alpha^-$, $\alpha^+$) could simply be swapped. Because now $0 < \alpha^-(p_{j\sigma}) \leq \alpha^+(p_{j\sigma})$, the value of

$\alpha^-(p_{j\sigma})$ will be disregarded and only $\alpha^+(p_{j\sigma})$ will be used to find an optimum $\alpha$.






| $j$-sigma rule | $p_{j\sigma} = \Phi(j)$ | $2p_{j\sigma} - 1$ (% of data captured) | $\alpha^+(p_{j\sigma})$ (Gaussian) | $M(\alpha^+(p_{j\sigma}))$ (Pearson family) | Suggested Sample size |
|---|---|---|---|---|---|
| "$\pm 3\sigma$" | 0.99865 | 99.73% | $\alpha_{3\sigma}^{\mathcal{N}} = 1.7$ | $\alpha_{3\sigma}^{\mathcal{P}} = 3.8$ | $n_{3\sigma} \sim 10^2$ |
| "$\pm 4\sigma$" | 0.9999683 | 99.994% | $\alpha_{4\sigma}^{\mathcal{N}} = 2.5$ | $\alpha_{4\sigma}^{\mathcal{P}} = 6.7$ | $n_{4\sigma} \sim 10^4$ |
| "$\pm 5\sigma$" | 0.9999997 | 99.99994% | $\alpha_{5\sigma}^{\mathcal{N}} = 3.2$ | $\alpha_{5\sigma}^{\mathcal{P}} = 9.4$ | $n_{5\sigma} \sim 10^6$ |

**Table 1.** Values of $\alpha^+(p_{j\sigma})$ for $j = \{3,4,5\}$ associated with the Gaussian distribution (4th column, $\alpha_{j\sigma}^{\mathcal{N}}$) and the distributions from the Pearson Family (5th column, median $\alpha_{j\sigma}^{\mathcal{P}}$). $\Phi$ is the cumulative distribution of the Standard Normal distribution $\mathcal{N}(0,1)$ and M is the median. The sample sizes $n_{j\sigma}$ correspond to less than 1 erroneously flagged outlier (based on the percentage of data captured).

### 2.2.2 The Pearson family

Univariate datasets are represented in this study with 9702 distributions from the Pearson family (Pearson, 1895; 1901 & 1916) composed of the Gamma (196 distributions), Inverse gamma (170), Beta (4703), Beta prime (1135), Pearson IV (3377) and Student (120) distributions. Their quantile functions are already implemented in $R$ to compute $\alpha^+(p_{j\sigma})$, and their *shape* parameters have been chosen to produce regularly-spaced points in the $(\kappa_{ex}, S^2)$ space without overlap and with a mean distance of 0.05 between them (Fig. 1).

The range of excess kurtosis and squared skewness for all distributions has been picked as $\kappa_{ex} \in [0,6]$ and $S^2 \in [0, \frac{4}{5}(\kappa_{ex} + 2)]$. Platykurtic distributions ($\kappa_{ex} < 0$) are discarded because the Gaussian case ($\kappa_{ex} = 0$) is considered as a minimal limit. Although the Pearson inequality is valid for any distribution ($S^2 \leq \kappa_{ex} + 2$; Pearson, 1916), values are narrowed down to $S^2 \leq \frac{4}{5}(\kappa_{ex} + 2)$ to exclude unrealistically skewed distributions (maximum Medcouple of 0.9, see next section). Finally, the choice of $\kappa_{ex}^{max} = 6$ has been picked with regard to the Exponential distribution ($S = 2, \kappa_{ex} = 6$), commonly used to delimit the realm of heavy-tailed distributions (such as Weibull or Fréchet).

### 2.2.3 Models

The general procedure developed in this study (*LogBox*) is based on the boxplot rule and replaces the original constant $\alpha = 1.5$ with $\alpha = k \log(n) + 1$, with $n$ the sample size and $k = 0.6$ the default value. This relationship is established based on the median values $\alpha_{j\sigma}^{\mathcal{P}} = \{3.8, 6.7, 9.4\}$ from the Pearson family, and the three sample sizes $n_{j\sigma} = \{10^2, 10^4, 10^6\}$ that correspond on average to less than 1 erroneously flagged outlier (see Table 1 & Fig. 1). *LogBox* is compared with four other models





(Kimber, 1990; Hubert & Vandervieren, 2008; Schwertman et al., 2004; Leys et al., 2013). The first two models (*Kim.* and *Hub.*) adjust the boxplot method with respect to the skewness:

$$\begin{cases} l_{Kim.} = q(0.25) - 3 \times \big(q(0.50) - q(0.25)\big) \\ u_{Kim.} = q(0.75) + 3 \times \big(q(0.75) - q(0.50)\big) \end{cases}$$

And

$$\begin{cases} l_{Hub.} = q(0.25) - 1.5 \times f(-MC) \times \big(q(0.75) - q(0.25)\big) \\ u_{Hub.} = q(0.75) + 1.5 \times f(MC) \times \big(q(0.75) - q(0.25)\big) \end{cases}$$

With the function $f$ defined as $f(MC) = e^{4MC}$ for $MC < 0$ and $f(MC) = e^{3MC}$ for $MC \geq 0$. The Medcouple $MC \in [-1,1]$ is a robust estimator of $S$, with a breakdown point of 25% and an algorithm complexity of $O(n \log n)$. The structure, power and reliability of this estimator are detailed in Brys et al., 2004. The third model (*Sch.*) constructs the lower and upper boundary around the median:

$$\begin{cases} l_{Sch.} = q(0.50) - \dfrac{Z}{k_n} \times 2\big(q(0.50) - q(0.25)\big) \\ u_{Sch.} = q(0.50) + \dfrac{Z}{k_n} \times 2\big(q(0.75) - q(0.50)\big) \end{cases}$$

With $k_n$ a function of the sample size $n$ to adjust for small samples (given as a table in Schwertman et al., 2004) and $Z$ a constant related to the percentage of data captured by $[l_{Sch.}, u_{Sch.}]$. The value of $Z = 3$ has been picked because it corresponds to the Gaussian case for the $\pm 3\sigma$ window (Table 1).

Finally, the last model (*Ley.*) uses a robust approximation of the standard deviation called the Median Absolute Deviation ($MAD$) that is defined as $MAD = 1.4826 \times M(|x - M(x)|)$ with M the median operator. The boundaries are expressed around

the median value as well:

$$\begin{cases} l_{Ley.} = q(0.50) - 3 \times MAD \\ u_{Ley.} = q(0.50) + 3 \times MAD \end{cases}$$

### 2.2.4 Comparison between models

The comparison between models is performed on a subset of the Pearson Family (600 distributions, supplementary material).

This subset has been created to give the same weight to each type of distribution, with 100 random distributions from the Beta, Betaprime, Pearson IV, Student, Gamma and Inverse-Gamma. Otherwise, the comparison would have had a large bias toward the Beta and Pearson IV (~83% of the Pearson family together). For a given model, a given distribution and a given sample size $n$, the following procedure is performed to calculate the percentage of data captured by the model (with $m = 0$ initially):

Step 1: generate random deviates (of sample size $n$)

Step 2: calculate $l$ and $u$

Step 3: Let $m_{within}$ be the number of points falling within $[l, u]$





$$m \leftarrow m + m_{within}$$

Step 4: Repeat $N$ times Step 1 to Step 3, with $N = ceiling\left(\frac{10^6}{n}\right)$

Finally, the percentage of data captured by the model for the given distribution and given sample size is $\frac{m}{(N+1)n}$. This percentage

is computed for all of the distributions, and the median value of the population of 600 percentages is defined as $M_i$, associated

with a sample size $n_i = 2^i$ varying from $n_4 = 16$ to $n_{14} = 16384$ (Fig. 2).

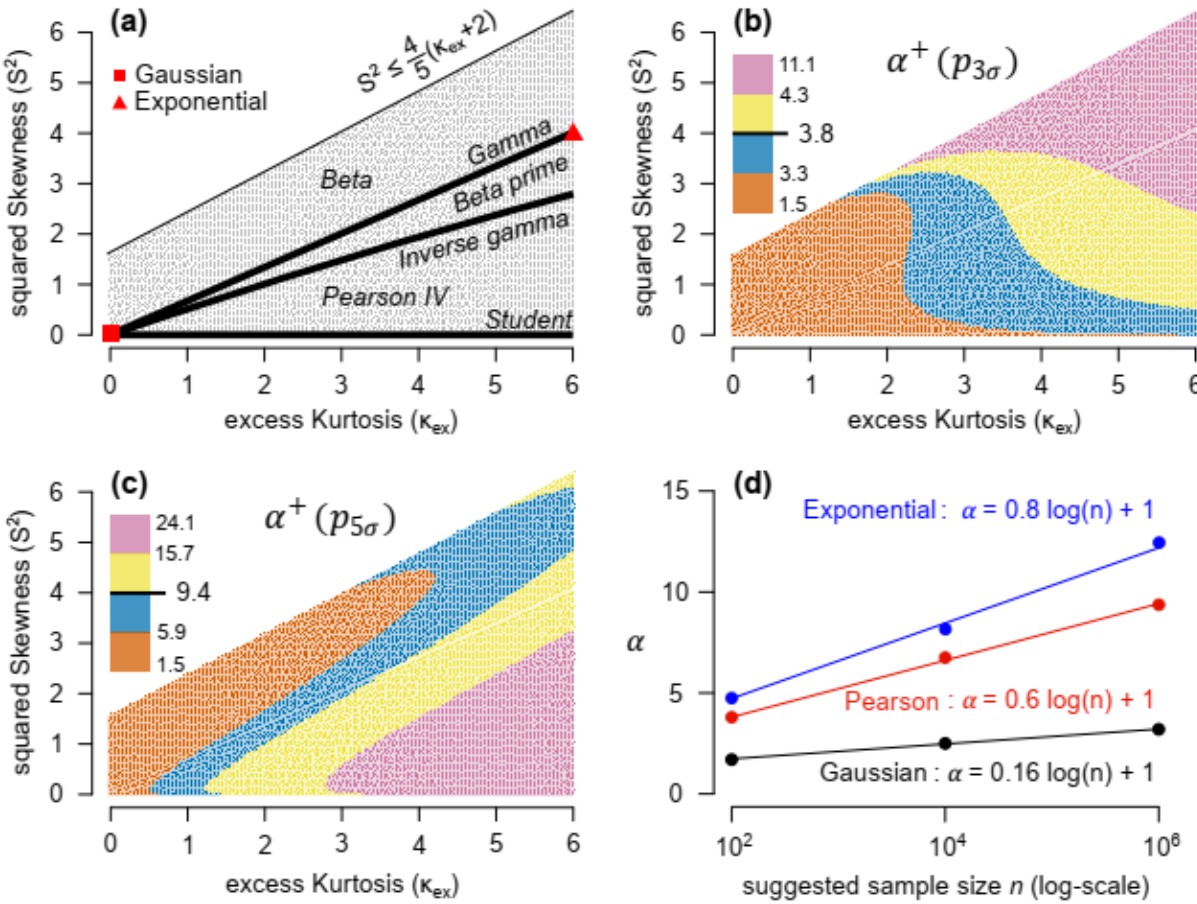

**Fig 1.** Location of the 9702 distributions of the Pearson family in the $(\kappa_{ex}, S^2)$ space (panel a). Values of $\alpha^+(p_{j\sigma})$ are shown
for $j = 3$ (**99.73%** of data captured, panel b) and $j = 5$ (**99.99994%**, panel c). The five numbers indicated in the legend
are the minimum, the three quartiles and the maximum. Relationships ($R^2 = 0.99$) between $\alpha^+(p_{j\sigma})$ and the sample size $n_{j\sigma}$
for the Gaussian, Exponential and Pearson family (panel d, see also Table 1).





### 170    2.3 Results and discussion

The original boxplot rule captures 99.3% of a Gaussian population using the constant $\alpha = 1.5$. In Table 1, this constant is shown to take larger values when considering wider windows (above 99.3%) and non-Gaussian distributions (Pearson family, Fig.1a). In the Gaussian case, $\alpha$ ranges from $\alpha_{3\sigma}^{\mathcal{N}} = 1.7$ to $\alpha_{5\sigma}^{\mathcal{N}} = 3.2$. Both values are similar to the $\alpha = 1.5$ and $\alpha = 3$ originally used by Tukey (1977) to describe "outside" and "far out" outliers. However, for non-Gaussian data with moderate

skewness and excess kurtosis, these criterions are too restrictive. The median of the $\alpha^{+}(p_{j\sigma})$ values from the Pearson family ranges from $\alpha_{3\sigma}^{\mathcal{P}} = 3.8$ (therefore above $\alpha_{5\sigma}^{\mathcal{N}}$ !) to $\alpha_{5\sigma}^{\mathcal{P}} = 9.4$. The three sample sizes given in Table 1 correspond on average to less than 1 data point erroneously flagged as an outlier, and lead to the relationship $\alpha = k \log(n) + 1$ with $k = 0.16$ in the Gaussian case, $k = 0.8$ in the Exponential case and $k = 0.6$ for the Pearson family (Fig. 1d).

The difference in $\alpha$ values among the distributions of the Pearson family is interesting to visualize in the $(\kappa_{ex}, S^2)$ space (Fig.

1). A non-linear relationship is observed between $\alpha$ and $(\kappa_{ex}, S^2)$, and the direction of this relationship depends on the outlier threshold. For the $\pm 3\sigma$ convention (Fig 1b), $\alpha$ increases with both $\kappa_{ex}$ and $S^2$. For the $\pm 5\sigma$ convention (Fig 1c), $\alpha$ increases with $\kappa_{ex}$ but decreases with $S^2$, which shows that a higher skewness does not necessary lead to more extreme events. While the Pearson family is ubiquitous in nature, one could argue that other systems exist. Different families of distributions were investigated (Jones, 2015), but they either did not cover the entire $(\kappa_{ex}, S^2)$ space, or no expression of $\kappa_{ex}, S^2$ or $Q$ (quantile

function) were available in a closed-form. The Generalized Lambda Distribution (GLD) system has been used in a previous study (Carling, 2000), but all distributions from this family have a bounded support and they therefore produce unrealistic datasets ($\alpha_{3\sigma} = \alpha_{4\sigma} = \alpha_{5\sigma} = 11$, see supplementary material).

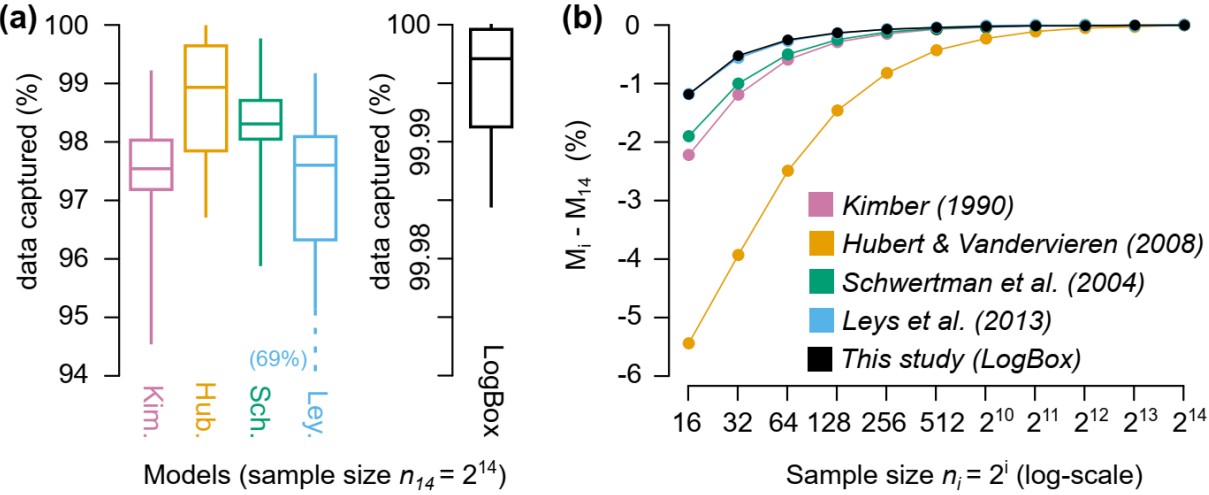

**Fig. 2.** Comparison between the five models performed on a subset of 600 distributions from the Pearson family (100 random distributions from the Gamma, Inverse gamma, Beta, Betaprime, Pearson IV and Student). Percentage of data captured by each model for a large sample size ($\boldsymbol{n_{14} = 2^{14} = 16384}$, panel a). Deviation of the median value $\boldsymbol{M_i}$ (associated with a





sample size $n_i = 2^i$) from $M_{14}$ (panel b). The boxplots show 95% of the 600 distributions (the whiskers are the 2.5% and 97.5% quantiles).


The comparison between models is described in the following. For a purely Gaussian distribution, *Kim.* and *Hub.* theoretically capture 99.3% of the data (boxplot rule) while *Sch.* and *Ley.* capture 99.73% ($\pm 3\sigma$ convention). For non-Gaussian distributions with a large sample size, these values are found to be lower: 97.5%, 98.9%, 98.3%, and 97.6%, respectively

(Fig. 2a). These four models therefore produce a high number of erroneously flagged outliers for any arbitrary distributions: around 150 extreme events are cut on average for 1 year of hourly data. This is not the case of *LogBox* that captures 99.997% of data, which corresponds on average to less than 1 outlier flagged in the sample size of $2^{14}$.

Models also show different sensitivity to the type of distribution encountered (error bars in Fig. 2a). The most stable model is *LogBox* (95% of the distributions fall between 99.98% and 100% of data captured), while *Ley.* is the most sensitive with a

percentage of data captured varying from 69.3% to 99.2%. This poor performance can be explained by the use of *MAD*, which contains a scaling factor parametrized on the standard deviation of the Gaussian (Leys et al., 2013). Quantile-based models (*Kim.*, *Sch.* & *Hub.*) are therefore more reliable for non-Gaussian distributions and show a variation of only $\pm 2\%$ around their median value.

Finally, the sensitivity of the models to the sample size is tested (Fig. 2b). All models show a negative bias in the percentage

of data captured compared to the large sample size. This bias is minimal with *LogBox* ($-1.2\%$) but important with *Hub.* ($-5.4\%$ for $n = 16$). This comes from the complexity of their model: the Medcouple is a remarkable estimator of the skewness but it requires a large sample size to reach convergence. In parallel, the protocol established by Schwertman et al. (2004) to correct for a small sample size effect has not shown significant improvements compared to other models (*Kim.* or *Leys.*).

To summarize, *Logbox* is a simple method inspired by the Boxplot rule to flag outliers in univariate datasets. It captures a

percentage of data that is adapted to large sample sizes and the k input offers flexibility in terms of outlier cutoff. The outlier level is drastically different between a Gaussian (k = 0.16) and Exponential distribution (k = 0.8), and this study suggests a default value of k = 0.6 based on distributions with moderate skewness and kurtosis (Pearson family). *Logbox* is implemented in the aggregation procedure that will be described in part II.



## 3 Part II, the past procedure

### 3.1 Context

This second part is dedicated to the pre-processing, partial imputation and aggregation of univariate time series. While aggregating data without outliers or missing values is a trivial task, it becomes more difficult when the data are contaminated. In order to flag outliers in a time series, one first needs to produce residuals that represent the variability around the *signal*. In its simplest form, the time series $y_t$ is represented with the following additive decomposition (Hyndman & Athanasopoulos, 2018): $y_t = T_t + S_t + \epsilon_t$, with $T_t$ a long-term trend, $S_t$ a cyclic component (originally, *seasonal component* but the term cyclic is preferred here as it is more generic) with period $\tau$ ($\forall\, t,\, S_t = S_{t+\tau}$) and $\varepsilon_t$ the remainders or residuals. One popular algorithm that performs this decomposition is the Seasonal and Trend decomposition using Loess (or STL, Cleveland et al., 1990), that is robust to the presence of outliers. The enhanced version of the algorithm, STLplus (Hafen, 2016), is also robust to the presence of missing values and data gaps. Unfortunately, there are three major drawbacks to using STLplus in the general case: (i) This algorithm has specifically been designed for signals showing seasonal patterns, which makes it less relevant for other types of data; (ii) The long-term trend based on loess needs to be parametrized. The decomposition is therefore not unique and the parametrization of the loess might seem arbitrary; (iii) The algorithm has a complexity of $O(n^2)$ due to the loess, which is resource intensive and not adapted to long time series ($n > 10^7$).

A new robust and nonparametric procedure (*past*) is proposed instead to calculate $T_t$ and $S_t$ using non-overlapping bins. Outliers are then flagged in the residuals $\epsilon_t$ with the *LogBox* method described in part I, and imputation is performed using $T_t + S_t$ if the cyclic pattern is strong enough, which is quantified by a new index introduced in this study (the Stacked Cycles Index or SCI). Bins with sufficient data can finally be aggregated, while other bins are discarded. The procedure is simple (entirely described in Fig. 3), the long-term trend $T_t$ is unique and non-parametrized (based on linear interpolations crossing each bin), the cyclic component $S_t$ is simply the mean stack of bins using detrended data (equivalent to STL for periodic time series). The algorithm complexity is of the order of $O(n\, log(n))$ because the loess is not necessary anymore.

In the following, the procedure is first described more in details and then applied to three case studies (a temperature, precipitation and methane dataset) that have been contaminated with outliers, missing values and data gaps. Comparison with the raw data demonstrates the reliability of the *past* procedure.

### 3.2 Method

#### 3.2.1 Definitions

Bin: a *bin* is a time window characterized by a left *side* (inclusive), a right *side* (exclusive), a *center* and a *period* (e.g., 1 year in Fig. 3a). Any univariate time series can be decomposed in a sequence of non-overlapping bins, with the first and last data





point contained in the first and last bin, respectively (Fig. 3a). The *bin size* $n_{bin}$ is the rounded median of the number of data points (including NA values) present in each non-empty bin of the sequence. A bin is *accepted* when its number of non-NA data points is above $n_{bin}(1 - f_{NA})$ with $f_{NA} \in [0,1]$ the maximum fraction of NA values per bin (input left to the user). Otherwise, the bin is *rejected* and all its data points are set to NA (Fig. 3a, bin 4).

Long-term trend: the *long-term trend (median based)* is a linear interpolation of the median values associated with each side (calculated between two consecutive centers, see Fig. 3a). A side value is set as missing if the number of non-NA data points (between the two nearest consecutive centers) is below $n_{bin}(1 - f_{NA})$. To solve for boundaries issues and missing sides values, the interpolation is extended using the median value associated with each center (bin 1, 3 & 5 in Fig. 3a). Once the outliers have been quarantined, the *long-term trend (mean based)* will be calculated following the same method but using the mean

instead of the median (Fig. 3c).

       Cycle: the *cycle (median based)* is composed of $n_{bin}$ points that are the medians of the stack of all accepted bins with the long-term trend (median based) removed (Fig. 3.b.1). Once the outliers have been quarantined, the *cycle (mean based)* will be the mean stack of accepted bins with the long-term trend (mean based) removed (Fig. 4a; bin 2,3 & 5). The cyclic component $S_t$

is the sequence of consecutive cycles.



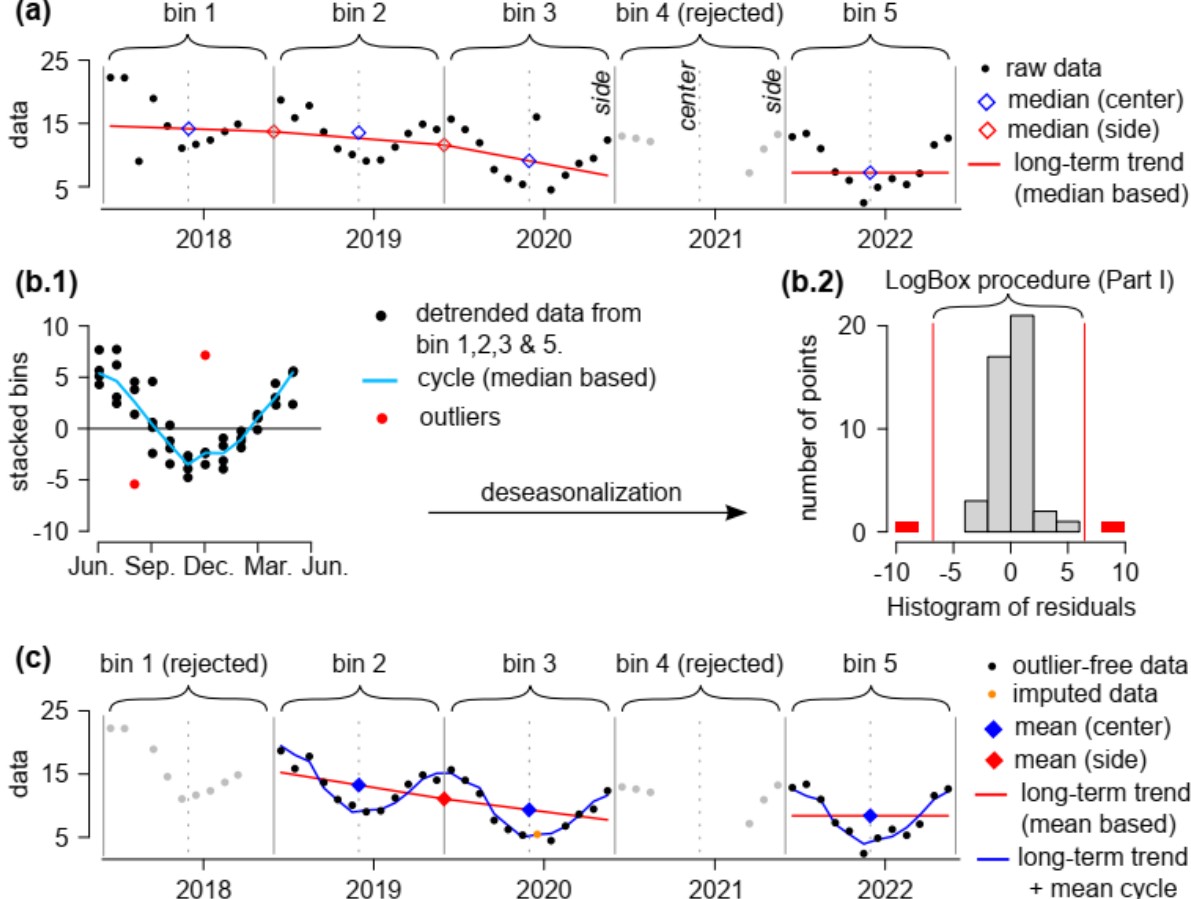

**Fig. 3**. Example of the aggregation procedure with the following inputs: bin side = 2020-06-01, bin period = 1 year, $f_{NA} = 0.2$ (minimum of 10 months of data for a bin to be accepted), $k = 0.6$ (outlier level) and $SCI_{min} = 0.6$ (cyclic imputation level). The bin 4 has been rejected because it contains only 6 months of data (panel **a**). Two outliers have been flagged in the residuals (detrended and deseasonalized data, panel **b.2**). After the outliers have been replaced with NA values, the bin 1 has been rejected (9 months of data), and the long-term trend and cycle have been updated using the mean instead of the median (panel **c**). A point in bin 3 has been imputed based on the cyclicity ($SCI_{min} \leq SCI = 0.61$ ).

Stacked Cycles Index:  SCI ≤ 1 is an adimensional parameter quantifying the strength of a cycle based on the variability around the mean stack (Fig. 4). Its structure is similar to another index developed in a former study (Wang et al., 2006), however a factor of $N_{bin}^{-1}$ has been added to correct for a bias emerging at a small number of bins ($N_{bin}$ is the number of accepted bins). This correcting factor has been calculated based on stationary time series of Gaussian noise (with therefore a null cyclicity per definition, see supplementary material).



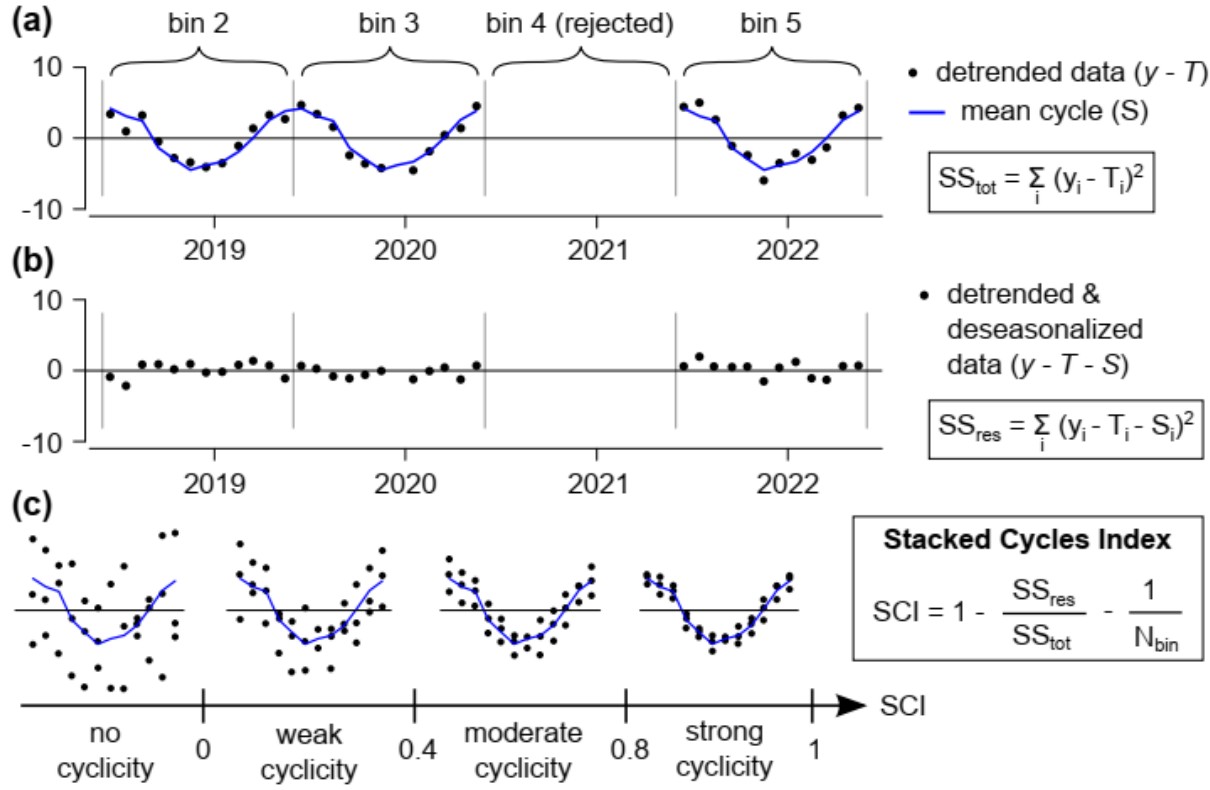

**Fig. 4**. The Stacked Cycles Index ($SCI \leq 1$) quantifies the strength of the cyclicity associated with the period of a bin (data from Fig. 3c). The long-term trend (mean based) is first removed to compute the total sum of squares (panel **a**). Then the cyclic component (mean based) is also removed to compute the sum of squared residuals (panel **b**). SCI is the coefficient of determination minus $N_{bin}^{-1}$ to correct for a bias emerging at a small number of bins, with $N_{bin}$ the number of accepted bins (here $N_{bin} = 3$, panel **c**).

### 3.2.2 *Past* procedure

Inputs.

1. The univariate time series ($1^{st}$ and $2^{nd}$ column: time and raw data, respectively)

2. One bin center or one bin side (e.g., 2020-06-01)

3. The period of the bin (e.g., 1 year)

4. The aggregation operator (mean, median or sum)

5. The range of possible values (default value $y_{lim} \in\, ]-\infty, +\infty[$)

6. The maximum fraction of NA values per bin (default value $f_{NA} = 0.2$)

7. The $k$ outlier level used in *LogBox* (default value $k = 0.6$)

8. The minimum SCI for imputation (default value $SCI_{min} = 0.6$)





Outputs.

1.  The raw dataset, with 8 columns: (i) time; (ii) outlier-free and imputed data; (iii) index of the bins associated with each data points (the index is negative if the bin is rejected); (iv) long-term trend; (v) cyclic component; (vi) quarantined outliers; (vii) value of the imputed data points; (viii) relative position of the data points in their bins, between 0 (the point falls on the left side) and 1 (the point falls on the right side)

       2.  The aggregated dataset, with 10 columns: (i) aggregated time (center of the bins); (ii) aggregated data; (iii) index of

305        the bin (negative value if the bin is rejected); (iv) start of the bin; (v) end of the bin; (vi) number of points per bin (including NA values); (vii) number of NA values per bin, originally; (viii) number of outliers per bin; (ix) number of imputed points per bin; (x) variability associated with the aggregation (standard deviation for the mean, MAD for the median and nothing for the sum)

       3.  The mean cycle, with 3 columns: (i) time boundary of the first bin with $n_{bin}$ points equally spaced; (ii) the mean

310        value associated with each point; (iii) the standard deviation associated with the mean value

       4.  The Stacked Cycle Index

       5.  representative number of data points per bin, $n_{bin}$

Step 1, data screening. The bin size $n_{bin}$ is calculated; values above or below $y_{lim}$ are set to NA; the number of accepted bins $N_{bin}$ is assessed; all data points within rejected bins are set to NA; the long-term trend and cycle (both median based) are

calculated (Fig. 3a,b.1).

Step 2, outliers. Outliers are flagged in the residuals (detrended and deseasonalized data) using the LogBox procedure (Fig. 3b.2); outliers are quarantined and their values are set to NA; the number of accepted bins $N_{bin}$ is updated; all data points within newly rejected bins are set to NA (Fig. 3c, bin 1).

Step 3, long-term trend and cycle (mean based): The long-term trend and the cycle are calculated using the mean instead of

the median (Fig. 3c); SCI is calculated (Fig. 4).

Step 4, imputation: If $SCI > SCI_{min}$, all NA values in **accepted** bins are imputed with the long-term trend + the mean cycle (imputation bounded by $y_{lim}$). Repeat Step 3 and Step 4 three times to reach convergence.

Step 5, aggregation: Accepted bins are aggregated around their center.

**3.2.3 Case studies**

Three univariate datasets are chosen to illustrate the potential of the aggregation procedure (Fig. 5, first column). The first dataset is an in-situ temperature (in °C) measured during summer in the canopy of an Oak woodland of California (month of August, temporal resolution of 5 min), and provided by the National Ecological Observatory Network (NEON, site SJER). The second dataset is an in-situ daily precipitation record (in mm) measured at the station of Cape-Leeuwin (South westerly

coast of Australia) from 1990 to 2020 and available on the Global Historical Climatology Network (Menne et al., 2012;





Xungang et al. 2012). The last dataset is a Methane proxy record (in ppbv) published in Loulergue et al. (2008) that covers 800000 years with irregular timesteps (varying from 1 to 3461 years, with a median of 311 years). None of the datasets contain obvious outliers or large data gap.

### 3.2.4 Contamination of the datasets

To test for the robustness of the aggregation procedure, the three raw datasets are contaminated by 30% (Fig 5, second column) with the use of three data gap (20% of the dataset), random NA values (9.5% of the dataset) and outliers (0.5% of the dataset). The three data gaps are picked with random length and position. The position of the outliers and the NA values follows a Poisson law. The value of the outliers is picked equal to $y_{min} - \frac{1}{2}(\mu - y_{min})$ or $y_{max} + \frac{1}{2}(y_{max} - \mu)$ with $y_{min}$, $y_{max}$ and $\mu$ respectively the minimum, maximum and mean of the dataset. No negative outliers are set for the precipitation because these values are impossible.

### 3.2.5 Aggregation of the datasets

Each dataset (raw and contaminated version) is consecutively aggregated twice (Fig. 6). The temperature dataset is aggregated (using the mean) every hour ($n_{bin} = 12$) and then every day ($n_{bin} = 24$). The precipitation dataset is aggregated (using the sum) every month ($n_{bin} = 31$) and then every year ($n_{bin} = 12$). The methane dataset is aggregated (using the mean) every 2000 years ($n_{bin} = 4$) and then every 20000 years ($n_{bin} = 10$). For each dataset, the mean cycle of the second level of aggregation is shown in Fig. 5 (second column).

The aggregation inputs are chosen as default values. The only exceptions are $k = \infty$ for the raw data (outliers are not checked), $f_{NA} = 1$ for the Methane dataset (bins with at least 1 non-NA data point are accepted due to the high irregularity in the sampling frequency) and $y_{lim} = [0, +\infty[$ for the precipitation dataset (negative precipitation are impossible). The number of false positive (real data points flagged as outliers) and false negative (real outliers that have not been flagged) are counted during the first level of aggregation (Table 2).




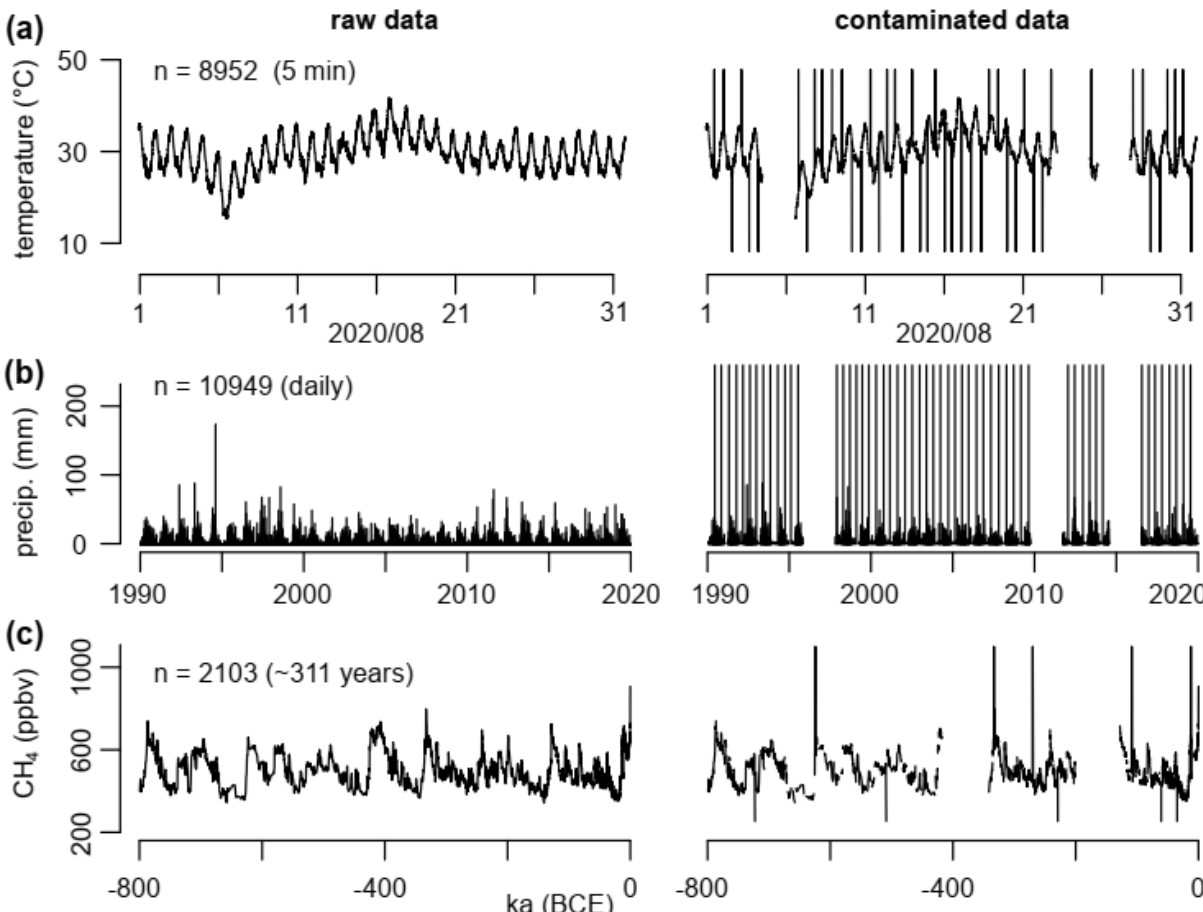

**Fig. 5**. Raw and contaminated versions of the three datasets used as case studies: temperature (panel **a**), precipitation (panel **b**) and methane (panel **c**). The sampling frequency is given in parenthesis. The contaminated versions contain three large data gaps (20% of the datasets), random missing values (9.5%) and random outliers (0.5%) set as a constant level below the minimum value and above the maximum value.

### 2.3 Results and discussion

The three univariate time series have been chosen as case studies due to their various statistical characteristics that are commonly seen in the scientific or economic field (Fig. 5, 1st column). The long-term trend follows smooth or moderate variations in the temperature and precipitation datasets, but shows a much higher volatility in the methane dataset. The cyclic pattern varies from strong diurnal cycles (temperature) and moderate seasonal cycles (precipitation) to no apparent cyclicity over a 20000 years period (methane). The detrended and deseasonalized residuals follow distributions from gaussian (temperature) or seemingly exponential (methane) to heavy-tailed (precipitation). Finally, the sampling frequency goes from sub-hourly (temperature) or daily (precipitation) to highly variable (1 to 3461 years, methane). To test the limits of the



aggregation procedure, these three datasets are *severely* contaminated by data gaps, outliers and missing values (Fig. 5, 2nd

column).



**Fig. 6**. Aggregation of the temperature (panel **a**), precipitation (panel **b**) and methane (panel **c**) in two consecutive levels: 1
(thin lines) and 2 (bold lines). Only the first level of aggregated precipitation is shown for clarity. Black and red colors are
associated with the raw and contaminated datasets, respectively. The mean cycles of the second level of aggregation are shown
in the second column, with their SCI displayed (the raw and contaminated versions share similar values).





The first level of aggregation recovers most of the destroyed signal with ~80% of the bins being accepted for all three datasets

(Fig. 6). In these accepted bins, all outliers have been correctly flagged (Table 2, zero false negatives). The mean percentage of difference between the contaminated and raw aggregates (level 1) is virtually zero for the temperature ($0 \pm 0.2\%$, 1 standard deviation), small for the methane ($-0.1 \pm 2\%$) but large for the precipitation ($-10 \pm 25\%$). This comes from the significant number of extreme precipitations events (85 days) that have been erroneously flagged as outliers (Table 2, false positives). Daily precipitation events are known to follow heavy tailed distributions (Wilks & Wilby, 1999), which is why the default

outlier level of $k = 0.6$ is insufficient here. A value of $k \sim 5$ is optimum in this case as it preserves the extreme events while cutting the outliers (the mean percentage of difference becomes $-0.1 \pm 16\%$). However, for daily precipitation with exceptional droughts followed by floods, the adequate outlier value can reach up to $k \sim 50$. For the Methane dataset, the 4 false positives (Table 2) come from the difficulty for the long-term trend to properly capture abrupt changes in $CH_4$ over few centuries. Again, this problem can be solved by increasing $k = 0.6$ to $k \sim 1$ without affecting the false negatives. To

summarize, the $k$ value is ultimately left to the user (based on her/his prior knowledge of the data) because there are no proper ways to estimate it systematically as the skewness and kurtosis of the residuals remain unknown (see part I).

| datasets | Temperature ($n = 8952$) | Precipitation ($n = 10949$) | Methane ($n = 2103$) |
|---|---|---|---|
| false positives | 1 | 85 | 4 |
| false negatives | 0 | 0 | 0 |

**Table 2.** Number of false positives (real data points flagged as outliers) and false negatives (outliers that have not been flagged) in the first level of the aggregation of the contaminated datasets shown in Fig. 6 using $k = 0.6$ in the LogBox method. For the

precipitation, $k = 5$ would put both the false positives and false negatives to 0.

The second level of aggregation has been performed to test for the cyclicity in the signal (Fig. 6, 2nd column) using the mean cycles and their associated Stacked Cycles Index (Fig. 4). The raw and contaminated mean cycles share similar magnitude within 1 standard deviation on the mean, and their SCI are the same: $-0.02$ for the methane (no apparent cycles of 20000

years period), 0.69 for the precipitation (moderate seasonality) and 0.88 for the temperature (strong diurnal cycles). The SCI reveals itself being useful when comparing signals of different nature or periodicities, which is not possible for seasonal indices that only focuses on one field (e.g., hydrology) or data format. (e.g., monthly) such as the seasonality index of Feng et al. (2013). Interestingly, the *past* procedure manages to recover the seasonality of the precipitation dataset despite cutting most of the extreme events (Fig. 6, b2). This result illustrates the fact that climatic models are able to capture the mean trend while

having difficulties to simulate exceptional events (Asadieh & Krakauer, 2015).





The cyclicity seen in the temperature and precipitation is strong enough to impute the missing data in all accepted bins, which further improves the reconstruction of the signal. For example, 11 months have been imputed in 9 different years for the precipitation dataset. Using $k = 0.6$, the mean percentage of difference with the raw data went for these years from $-24 \pm 10\%$ (without imputation) to $-20 \pm 5\%$ (with imputation). Because $SCI$ has a similar structure than a coefficient of determination, imputations based on high $SCI$ ($> 0.6$) are not detrimental to the signal, which is not the case of most linear interpolation. Again, the choice of performing or ignoring the imputation is left to the user with the input parameter $SCI_{min}$ that will be compared to $SCI$ (see method).

To summarize, the aggregation procedure is able to filter contaminated data by selecting bins with sufficient quality (input: $f_{NA}$) which are then cleaned from outliers (input: $k$). The cyclic pattern within each bin is evaluated ($SCI$) and missing data are imputed in accepted bins if the cyclicity is strong enough (input: $SCI_{min}$). A prior knowledge of the data is essential to correctly chose $f_{NA}$, $k$ and $SCI_{min}$ values, as these differ between datasets.

**3 Conclusion & recommendation (Part I & II)**

Although univariate timeseries are the simplest type of temporal data, this study reveals a lack of consensus in the literature on how to objectively isolate outliers from the signal especially in raw data of poor quality. In part I, a comparison between outlier detection methods for univariate datasets has shown that extreme events are too often flagged as outliers (type I error), especially in non-Gaussian populations with a large sample size. This led to a new method (called LogBox) that improves the boxplot rule by replacing $\alpha = 1.5$ with $\alpha = k \log(n) + 1$, with $n$ the sample size and $k$ left to the user (default value of 0.6). In part II, a pre-processing procedure called past (implemented in R) based on data binning has been proposed to clean, decompose and aggregate signals without assumption on their structure. The strength of the cyclic pattern within each bin is assessed with a novel and adimensional index (the Stacked Cycles Index or SCI) inspired by the coefficient of determination. Most of the signal can be retrieved from messy univariate time series with diverse statistical characteristics.

The past procedure has several limits that can be addressed by varying the inputs: period of the bin, maximum ratio of missing values per bin ($f_{NA}$), outlier level ($k$) and cyclic imputation level ($SCI_{min}$). It is recommended to pick the period of a bin so that it contains on average between 4 and ~50 data points. Below 4 would decrease the breakdown point to unsafe levels (1 outlier would be enough to contaminate the bin), and above 50 would produce a long-term trend that might not properly capture the variability in the signal. A maximum of 20% of the bin can be missing by default ($f_{NA} = 0.2$), but when data are sparse and irregularly distributed, a value of $f_{NA} = 1$ is possible (example of the Methane dataset: bins with only 1 data point were



accepted). An outlier level of $k = 0.6$ will minimize the type I and II errors in a majority of cases, but can vary up to $k = 50$ for time series with exceptional spikes (e.g., daily rain with a 11-months drought and then a 1-month flood). Finally, the imputation level (default of $SCI_{min} = 0.6$) can vary between 0 (forced imputation even without cyclic pattern) and 1 (no

imputation). It is strongly recommended to examine the data before and after using the procedure to ensure that rejected bins and flagged outliers seem reasonable, and to be transparent about the inputs used in your future study.

**Author contribution**

F.R.: Design, writing, coding.


**Competing interests**

The author declares no competing interests.

**Acknowledgement**

The author would like to warmly thank Rob Hyndman for his advice.

**Data availability**

The GHCN dataset is available on https://www1.ncdc.noaa.gov/pub/data/ghcn/daily/. The Methane dataset is available on https://doi.org/10.1038/nature06950. The temperature dataset is available on https://doi.org/10.48443/2nt3-wj42.


**Code availability**

The past package is currently unavailable on the comprehensive R Archive Network (CRAN) because the manuscript "Technical note: A procedure to clean, decompose and aggregate time series" is still under review. Please contact the author (ritter.francois@gmail.com) if you are interested in receiving the package separately.

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
