# Peer review of "Technical note: A procedure to clean, decompose and aggregate time series"

_Hydrology and Earth System Sciences, 2021_

## Author Comment (AC1)

Dear Anonymous Referee #1,

Thank you for your detailed feedback. I am sorry to learn that, overall, you did not find a particular interest in this work. I probably miscommunicated the importance of pre-processing, the improvement of the boxplot rule and the description of the strength of a cyclicity in a signal. I hope this response to your comments will clarify some points.

First of all, I would like to apologize concerning the package availability (I have had trouble uploading it on Github). It is now publicly available (https://github.com/fritte2/ctbi) as well as the code used in the study (https://github.com/fritte2/hess-2021-609). Due to name conflicts, the name **past** has been replaced with **ctbi**, which stands for "Cyclic-Trend decomposition using Bin Interpolation". It can be installed with the following command: "library(devtools) ; install_github("fritte2/ctbi") ; library(ctbi)".

In the following, the main points of your review (in grey) are highlighted in bold.

**i) The structure and aim of the manuscript do not fit the HESS journal**

One main concern is that the structure of the manuscript (length, aim, introduction, content) is not adapted to a HESS Technical note.

1) The title says it is a technical note, but the structure does not really fit that. According to instructions, submissions of technical notes should be only a few pages, while this submission is substantially longer. Also, in the abstract it is stated that the submission wants to propose a standardized way to pre-process time series, which is not really something that can be done in a technical note. However, to be a full research article several important parts are missing in the submission (see comments further down)

2) The article lacks references and introduction in the field of HESS. The introduction is quite general on time-series and R packages, but does not discuss what is commonly used in hydrology or earth sciences and to which extend there is a need for additional improvement of outlier detection and gap imputation within these areas. [...]

[…]

Potentially the submission could be resubmitted as a purely technical note describing the R package and discussing the possible inputs to the function (k, …) and with some examples of different choices on the output. Such a submission should more intuitively describe which effect a change on k has, rather than rely on a simulation study that is difficult to relate to in practice. For example, describing how it might work for a normal and a log-normal distribution often met in earth science.

I agree that the manuscript is difficult to put in a specific category. It does not discuss any physical mechanism encountered in Hydrology or even Earth Science (this rejects the manuscript as an Earth Science research paper); however, the package and the framework are dedicated to all researchers dealing with temporal measurements coming from Earth Science (this rejects the manuscript as a purely statistical paper). I decided to submit it to HESS (and not a statistical journal) because the need for this R package originally came from my field (hydrology & ecosystem), where in-situ or remote measurements are often of poor quality. Additionally, there is a clear gap between statisticians and Earth scientists, as these two separate communities do not read the same journals.

This manuscript was originally submitted as a research paper (which explains the length), but the editor suggested to change it as a technical note. I can adapt the introduction to add a

paragraph entirely dedicated to Earth Science protocols on how to deal with outliers or missing values, but this paragraph will be related to what was already described (e.g., boxplot rule, STL, Hampel filter...), or will be completely specific to a field (e.g., extreme value theory used in return event of 100-year floods). In any case, this will increase the length of the final article.

Concerning the potential resubmission as a purely technical note describing the package, the simulation based on the Pearson family is fundamental in the problem of flagging outliers and justifies the creation of the *Logbox* method. The logarithmic law of $\alpha = k \log(n) + 1$ is solely based on the Pearson family. I therefore disagree on cutting it.

The discussion about how the *LogBox* method works for a Normal distribution is already present in the manuscript (L65-66, L75, L95, L171-175 & Table 1).

**ii)     The framework developed in this manuscript is unnecessary**

Another concern is that, overall, the study does not focus on essential problems in Earth Science. Gaps, outliers, univariate timeseries have been handled by each community separately so far, why do we need change things?

3) Generally, in environmental and earth science time series can have a large variety of different structures and the questions to be investigated vary widely. Depending on which statistical analysis is to be made, the filling of gaps or outlier detection can be more or less important. For many approaches, outliers or gaps are not a crucial problems, but can be handled intrinsically. It is, thus, not obvious that a standardized way to preprocess is desirable. Obviously, when several series are within the same academic study they should be handled similarly, but no examples of this being a real problem at present is given. Also, in earth sciences there are few situations where only single time series need to be handled. Either there are several variables observed at the same time point, which can be used to identify if there is something wrong with the sample altogether one variable specifically, or there are nearby stations available that can be used to identify outliers or fill gaps.

Firstly, the R package developed in this study is not entirely dedicated to flagging outliers or imputing missing values. It can be used without flagging outliers (*k.outliers = Inf* in **ctbi**) or imputing data (*SCI.min = Inf*), but simply as a tool to aggregate data in a flexible manner and quantify the strength of a cyclic pattern in a generic way. These two problems are non-trivial, they constitute half of the manuscript (part II), and they are not mentioned in your review.

Secondly, any discipline that produces measurements (in-situ or remote) has to handle gaps and outliers. Even the most standardized products (GHCN, MODIS, NEON,..) contain spikes or suspicious values. Before the creation of this package, I had no tools on how to properly preprocess them before performing the analysis. For example, the *tsoutliers* function (R package *forecast*) fails when data are non-gaussian. I therefore had to pre-process my raw data "manually", by removing errors "visually", in a procedure that is case-specific. This method is not only time-consuming; it is completely arbitrary. Two scientists pre-processing the same raw data would end up with two different preprocessed datasets. Additionally, the pre-processing method detailed in published articles is sometimes cryptic to anyone who did not manipulate the raw data.

I hope these points prove how much the non-standardization of preprocessing goes against the scientific rule of reproducibility. I have difficulties to understand how you never encountered

these issues when dealing with measurements, and how this problem can be considered as '*not crucial*'.

As you mentioned it, the procedure developed in the manuscript concerns univariate timeseries for a single location, and not multivariate timeseries recorded at different spatial locations. However, the procedure can be applied to each variable individually (with their own parametrization) to merge them into a standardized aggregated dataset. For example, if someone measures climatic variables (precipitation, temperature, humidity, …) at a different time resolution, **ctbi** will be able to aggregate them at the same resolution. The copyleft license (GPL-3) and the open-source nature of the package leave room for future improvements, for example spatial considerations.

**iii) The comparison with existing methods has not been performed**

2) [...]The only part that connects to the journal are the case studies, which are relevant, but as the results are not compared with other approaches or articles that have used these series before in a different way conclusions are difficult to make.

[…]

- At least one of the case studies has a seasonal pattern, which would allow a comparison to STL or Stlplus

[…]

- References to the use of outlier detection methods in earth sciences are missing.

The second part of the manuscript did not include any comparison with existing methods because I did not find a popular referenced method (or R package) that can handle the three contaminated cases together as they are so different from a statistical point of view (temperature, precipitation & methane). **Stlplus** does not handle the temperature and methane datasets, and the function **hampel** that applies a Hampel filter to flag outliers does not work on data with missing values. The function **tsoutlier** can be applied to the three datasets but it will cut too many data points (Gaussian assumption). I could compare my results to it, but, again, the article will be significantly longer and the goal of the article is not an inter-comparison of outlier detection methods.

Additionally, I have indicated L240 that the cyclic component of **ctbi** is identical to the cyclic component of **stlplus** for periodic time series. If someone is interested in comparing both decompositions, I can show the long-term trend of **stlplus** and the long-term trend of **ctbi** in a single graphic in the supplementary material.

- It is argumented that Stlplus has severe disadvantages compared to the proposed method. For example, it is said that the trend modelled with loess needs to be parametrized. No reference is given and it is unclear what is meant by this, as loess is a non-parametric regression methods and does not need a parametrization.

**stlplus** is a non-parametric method: it does not assume a particular distribution of the raw data. However, there are several input parameters that will change the shape of the LOESS. Namely, s.degree; t.window; t.degree; fc.degree; l.window; l.degree in the function **stlplus** of the R package **stlplus**. I can rephrase this sentence to clarify the difference between a non-parametric method and the input parameters required to use it.

**iv) The treatment of outliers is unclear**

**iv.a) Definition of an outlier**

- It is not discussed which definition of outlier is used in this context, and especially it would be important to define outliers in highly skewed distributions and how it would be possible to distinguish them from observations that belong to the distribution.

The definition of an outlier is established L58-62 within the framework of the boxplot rule: a value below a lower boundary $l$ or above a higher boundary $u$ is considered as an outlier. These two boundaries only depend on the value of $\alpha$, originally chosen as $\alpha = 1.5$ to fit the Gaussian distribution for a small sample size ($n < 100$).

I have adapted this framework to *any* kind of distribution (with an arbitrary skewness or kurtosis) and *any* sample size to define the function $\alpha^+$ that will replace $\alpha$ (see L86-87). The value of $\alpha^+$ depends on two parameters: 1) $Q$, the quantile function of the distribution that depends on the nature of the distribution (Gaussian, Exponential, etc.) and 2) $p$ that determines the percentage of data captured within $[l,u]$ (Table 1 columns 2 & 3).

A percentage of data captured is associated with a specific sample size (e.g., 99.73% of data captured is associated with a sample size of 100), see Table 1 columns 3 & 6. This will ensure that less than 1 outlier is flagged on average. This association led to the *Logbox* rule $\alpha^+(n) = k \log(n) + 1$, with $k$ depending on the nature of the distribution (see Fig. 1d).

**iv.b) The Pearson family**

- It is also not clear how the 9702 distributions are defined and how they are chosen. At one place, a reference to the supplementary is given, but there is no info on distributions in the supplementary.

The Pearson family is a system of distributions (Fig. 1a) that covers most of the kurtosis-skewness space. Each distribution has a one or several shape parameters that will change the value of the skewness and the kurtosis. L115: "their *shape* parameters have been chosen to produce regularly-spaced points in the ($\kappa$, $S^2$) space without overlap and with a mean distance of 0.05 between them (Fig. 1).". The value of the shape parameters is available on GitHub, these distributions are well-known (gamma, beta, beta-prime, student…) and I thought it was not necessary to detail all their characteristics.

Let's consider you have detrended & deseasonalized your data to extract the residuals that follow an unknown distribution. You will not be able to calculate an estimate of the kurtosis and the skewness of this distribution (it would require an extremely high sample size and no outliers). For the same reason, the $\alpha$ value cannot be estimated based on this distribution either (although the 4 models in Fig. 2 have attempted - and failed - to do so). The reasoning is the following: if the unknown distribution is *slightly* non-gaussian (moderate skewness & kurtosis), it will be one of the distributions of the Pearson family (one point in the cloud of distributions in Fig. 1a). And the most representative $\alpha$ of this family is the median of the 9702 distributions.

- It is rather unclear how the value of k are determined. Are simulations in Figure 1a-1c made for several sample sizes and their medians are shown in panel d?

Fig. 1b,c does not include any sample size. Based on the definition of $\alpha^+$ (L 87), you simply need the quantile function of a distribution (which is an analytical formula) and the value of $p$ (given in Table 1) to calculate $\alpha^+(p)$. The values of $\alpha^+(p)$ in Fig. 1b,c are therefore *exact*, they have not been numerically determined with simulations. The median is calculated on the population of all $\alpha^+(p)$ of the 9702 distributions of the Pearson family for a given $p$.

How do I find the law of $\alpha = k \log(n) + 1$ ? This law works the Gaussian (Table 1, column 4 versus column 6, $k = 0.16$), but also the medians of all $\alpha$ values of the Pearson family (Table 1, column 5 versus column 6, $k = 0.6$), as well as for the Exponential (Fig. 1d, $k = 0.8$).

- For comparison between outlier detection method 600 distributions were selected to give the same weight to different types of distributions. For determining the value of alpha and k all 9702 distribution are used. It is unclear why.
- It is rather unclear how well the suggested values of 3.8 and 9.4 work in practice as they are the median of values achieved in the simulation. This means probably that these values work considerably worse for some specific distribution. No discussion is made about this.

Why introducing 600 random distributions, 100 per family, to compare models? Two reasons: (i) If I randomly pick into the 9702 distributions with equal weight, I will have 83% of chance to pick a Beta or Pearson IV (L151), and the comparison will be biased. For example, the other models (Ley., Sch., Hub., Kim.) might perform much better on a student or gamma distribution. (ii) *Logbox* has been parametrized on these 9702 distributions, it would be incorrect to use the exact same pool of distributions to compare models.

The performance of the *Logbox* method on distributions with $\alpha$ values different from 3.8 or 9.4 (For example, orange and pink areas in Fig. 1b and Fig. 1c) is shown in Fig. 2a. These distributions contribute to the variability seen in the percentage of data captured by *Logbox*, which shows an excellent performance to distributions with various skewness & kurtosis (L203-204).

**iv.c) Type I error and type II errors are unclear**

- A new boxplot rule is suggested and motivated by that using this rule leads to far less false positives, i.e. the type I error is improved. No mention is made on the type II error, which is typically increases, when the type I error decreases. Clearly, this is not easy to study as, in a univariate time series, only outliers above a certain threshold can be detected. [..]. In the recommendations it is stated that the value of k=0.6 will minimize the type I and type II errors, but it is very unclear how this determined and generally it is not possible to minimize type I and type II errors at the same time.

Yes, a sentence on the difference between type I error (real data points are flagged as outliers) and type II error (real outliers are missed) and how they are related to the concept of "percentage of data captured by a model" is missing in the part I of the manuscript. This will be updated. Actually, the framework developed in part I with the theoretical distributions of the Pearson family takes both the type I and type II errors into account. The *Logbox* has been parametrized to flag, on average, less than 1 point as an outlier for a distribution with moderate skewness & kurtosis. Short example: if 1000 data points are generated following an Exponential distribution and the Logbox method cuts 1 point (it captures 99.9% of the data), it means (in this single case) that the Error of type I is 1 and error of type II is 0 (because it will cut any outliers above this threshold). If the percentage of data captured was 100%, there would be an issue of type II error because the threshold might be too high. But more than

~80% of the distributions of the Pearson family are cut below 100% (Fig 2a), which means there rarely is an issue of type II error.

The only problem is that the detrended & deseasonalized residuals might follow a distribution drastically different from those used in the Pearson family. This is for example the case of the daily precipitation (heavy tailed), with a large number of false positives. Users therefore need to make an educated guess on the nature of their data, from Gaussian (k=0.16) through Exponential (k=0.8) to heavy tailed (k > 5). The default value of k=0.6 concerns distributions departing from the Gaussian, with moderate skewness and moderate kurtosis.

- In this study this threshold is chosen to be very high, leading probably to situations where few (real) outliers are detected. This is also one of the reasons outlier detection methods flag rather many observations as outlies, giving the user the possibility to doublecheck the correctness of those and keep the ones that seem reasonable
- In the case studies, outliers are introduced and can be identified with the proposed method, but the outliers are completely unrealistic and could be identified by visual inspection only, no advanced methods are needed.

The threshold of $k = 0.6$ could actually be considered as not high enough. As you can observe in Table 2, there are false positives for the three datasets, including the temperature whose residuals are supposed to be Gaussian. This proves how much the classical boxplot rule ($\alpha = 1.5$) is outdated, and why some authors use $\alpha = 3$ instead (package **tsoutliers**, see https://robjhyndman.com/hyndsight/tsoutliers/).

How can someone determine if an outlier is "correct" once it is flagged? Is it through a visual inspection? More generally, it is difficult to say if a value is "realistic" based on "visual inspection only". In Earth Science, record breaking events (mega-droughts, mega-floods, etc..) are actually not realistic at all, which is exactly why models fail to capture them. A recent example: 202 mm of rain have fallen in 1 hour in Zhengzhou (China) on July 20th 2021. Without contextualization, a "visual inspection" would classify this event as impossible.

The level of outliers in part II has been chosen to illustrate the robustness of the method. If this level was too close to the raw signal, one could argue that these are not outliers, simply extreme events.

- The breakdown points of the outlier detection methods are not given.

I can provide the breakdown points, but they apply to the residuals that are not shown in the study.

- In many cases in environmental and earth science it is common to work with e.g. log-transformed values (or models that use a log-link) to account for single high values in skewed distributions. This would make it unnecessary to develop outlier detection methods for skewed distribution. Instead, conventional methods can be used on the symmetric transformed values. Has typical handling of skewed distributions in earth science been studied? Is there a need to find outliers in skewed distributions?

The log-transformation of data can be used in different contexts, such as producing Gaussian residuals (Cox-Box method) or extracting the different components of a signal in a particular case, from $y_t = S_t \times T_t \times \varepsilon_t$ to $\log(y_t) = \log(S_t) + \log(T_t) + \log(\varepsilon_t)$. A famous example is the AirPassengers data in R. In any case, these methods are case-specific and hazardous to

implement in a generic case. Also, a high *skewness* is not necessary a problem, but more often a high *kurtosis* leads to extreme events that seem impossible.

Comments on structure

- Referencing within the article is not clear. Often Figures are referenced already in the methods description, making the reading difficult. A better separation of method and results would be helpful
- Also sections called context and method are not clearly divided.

Can you please give me the specific lines and figures where these issues appear?

Thank you!

---

## Author Response (AR1)

Dear reviewers and dear editor,

Thank you for your feedbacks that were complementary. Major revisions of the manuscript have been made to take into account your suggestions, as well as to respect the structure and size of a "Technical note". It was decided to keep part I within the study to give elements of context to future readers (*LogBox* method), but also to shorten it as HESS is not a purely statistical journal.

In the following, you will find a short summary of the changes, then a point-by-point response to your suggestions. Most of these points already received a long individual response (interactive discussion), so in this note I simply pointed out the changes in the manuscript that correspond to specific issues.

Update summary:

**In the R package:**

1) **past** has been replaced with **ctbi,** due to name conflict. This package is currently under review on CRAN.
2) Some minor updates in the **ctbi** R package have slightly changed the number of false positives/false negatives in Table 2, these do not impact the figures, discussion or conclusion in the study. The updates in the **ctbi** are:
(i) $n_{bin\ min} = ceiling((1 - f_{NA}) \times n_{bin})$ instead of $n_{bin\ min} = floor((1 - f_{NA}) \times n_{bin})$ with $n_{bin\ min}$ the minimum number of points for a bin to be accepted ($n_{bin\ min}$ is now rounded up instead of rounded down, otherwise it was counter-intuitive).
(ii) The left side of the first bin and the right side of the last bin of a time series were treated as NA values, this has been removed.
3) The output $n_{bin\ min}$ is now available, otherwise the user did not know the minimum number of points for a bin to be accepted.

**In the manuscript:**
1) The *tsoutliers* pre-processing alternative has now an important role. It has recently been updated on CRAN (https://robjhyndman.com/hyndsight/tsoutliers/) to include a smoothing function for non-seasonal time series and a Cox-Box method to transform the residuals into a Gaussian (mentioned by Reviewer #1). *tsoutliers* is now better described in the introduction, and the false positives/negatives can be compared with **ctbi** in Table 2, as well as in the discussion. However, the long-term trend and cyclic component of *tsoutliers* are not available, which limits the comparison.
2) Part I has been shortened: most of the unnecessary elements of the method have been moved to the supplementary material.
3) The limits of **ctbi** concerning signals of residuals with non-stationary variance has been recognized in a new section: **Limits & recommendations.** These signals can be handled following a protocol applied to the soil respiration dataset MIGLIAVACCA of reviewer #2 in the supplementary material.
4) The importance of expert-knowledge in pre-processing has been recognized, particularly for periods of instrument failure or human errors where most algorithms usually fail.

Step by step response:

**Reviewer #1**

1) The title says it is a technical note, but the structure does not really fit that. According to instructions, submissions of technical notes should be only a few pages, while this submission is substantially longer. Also, in the abstract it is stated that the submission wants to propose a standardized way to pre-process time series, which is not really something that can be done in a technical note. However, to be a full research article several important parts are missing in the submission (see comments further down)

- The format of the article has been shortened to fit as much as possible the structure of a technical note. The use of $\alpha^+$ instead of $\alpha^-$, the choice of the boundaries of the Pearson family, the choice of the subset of 600 random distributions, the Generalized Lambda Distribution (GLD) system and their explanations have been moved to the supplementary material. The wording of the discussion has been simplified as well.

2) The article lacks references and introduction in the field of HESS. The introduction is quite general on time-series and R packages, but does not discuss what is commonly used in hydrology or earth sciences and to which extend there is a need for additional improvement of outlier detection and gap imputation within these areas. The only part that connects to the journal are the case studies, which are relevant, but

as the results are not compared with other approaches or articles that have used these series before in a different way conclusions are difficult to make.

[…]

References to the use of outlier detection methods in earth sciences are missing.

- the *tsoutliers* function is now mentioned in the introduction (L35-41), detailed in the method (L335-338), its performance is available in Table 2, and the discussion includes a comparison with **ctbi** (L373-377). This is the most general pre-processing option present in the field of HESS (*tsoutliers* is part of the package **forecast** which is extensively used for time series analysis in Earth Science).

I have also mentioned common problems related to measurements in Earth Science (L23-28) in response to Reviewer #2, but I did not dive into details as this is not the goal of this study.

3) Generally, in environmental and earth science time series can have a large variety of different structures and the questions to be investigated vary widely. Depending on which statistical analysis is to be made, the filling of gaps or outlier detection can be more or less important. For many approaches, outliers or gaps are not a crucial problems, but can be handled intrinsically. It is, thus, not obvious that a standardized way to preprocess is desirable. Obviously, when several series are within the same academic study they should be handled similarly, but no examples of this being a real problem at present is given. Also, in earth sciences there are few situations where only single time series need to be handled. Either there are several variables observed at the same time point, which can be used to identify if there is something wrong with the sample altogether one variable specifically, or there are nearby stations available that can be used to identify outliers or fill gaps.

Potentially the submission could be resubmitted as a purely technical note describing the R package and discussing the possible inputs to the function (k, …) and with some examples of different choices on the output. Such a submission should more intuitively describe which effect a change on k has, rather than rely on a simulation study that is difficult to relate to in practice. For example, describing how it might work for a normal and a log-normal distribution often met in earth science.

- Responses to these concerns have been given in my individual comment & the editorial decision.

A new boxplot rule is suggested and motivated by that using this rule leads to far less false positives, i.e. the type I error is improved. No mention is made on the type II error, which is typically increases, when the type I error decreases. Clearly, this is not easy to study as, in a univariate time series, only outliers above a certain threshold can be detected. In this study this threshold is chosen to be very high, leading probably to situations where few (real) outliers are detected. This is also one of the reasons outlier detection methods flag many observations as outlies, giving the user the possibility to doublecheck the correctness of those and keep the ones that seem reasonable. In the recommendations it is stated that the value of k=0.6 will minimize the type I and type II errors, but it is very unclear how this determined and generally it is not possible to minimize type I and type II errors at the same time.

[…]

In the case studies, outliers are introduced and can be identified with the proposed method, but the outliers are completely unrealistic and could be identified by visual inspection only, no advanced methods are needed.

- To avoid redundancy, I have removed the terms type I and type II errors and have replaced them with false positives and false negatives throughout the study. Two concerns were that the cutting threshold of *LogBox* is too high, and that the outliers used in part II were unrealistic. The comparison with *tsoutliers* shows that it is not the case:

    i. *tsoutliers* flags false negatives in all three datasets, proving that the outlier level is not unrealistic.

    ii. *LogBox* does not flag any false negatives, showing that the threshold of *LogBox* is not too high.

It is not discussed which definition of outlier is used in this context, and especially it would be important to define outliers in highly skewed distributions and how it would be possible to distinguish them from observations that belong to the distribution.

The breakdown points of the outlier detection methods are not given.

It is rather unclear how well the suggested values of 3.8 and 9.4 work in practice as they are the median of values achieved in the simulation. This means probably that these values work considerably worse for some specific distribution. No discussion is made about this.

It is rather unclear how the value of k are determined. Are simulations in Figure 1a-1c made for several sample sizes and their medians are shown in panel d?

For comparison between outlier detection method 600 distributions were selected to give the same weight to different types of distributions. For determining the value of alpha and k all 9702 distribution are used. It is unclear why.

It is also not clear how the 9702 distributions are defined and how they are chosen. At one place, a reference to the supplementary is given, but there is no info on distributions in the supplementary.

- Responses to these concerns have been given in my individual comments. I have given more details in the supplementary material that justify the choice of the 600 distributions, and why $\alpha^+$ is computed on the 9702 distributions instead of the subset.

In many cases in environmental and earth science it is common to work with e.g. log-transformed values (or models that use a log-link) to account for single high values in skewed distributions. This would make it unnecessary to develop outlier detection methods for skewed distribution. Instead, conventional methods can be used on the symmetric transformed values. Has typical handling of skewed distributions in earth science been studied? Is there a need to find outliers in skewed distributions?

[…]

At least one of the case studies has a seasonal pattern, which would allow a comparison to STL or STLplus

- The *tsoutliers* function gives the user the possibility to apply the Cox-Box method to the residuals, which transforms them into a Gaussian. Table 2 shows that this method does not properly work for heavy-tailed residuals, which makes the problem of flagging outliers non-trivial and also justifies this study. Additionally, the *tsoutliers* function uses the STL procedure.

It is argumented that STLplus has severe disadvantages compared to the proposed method. For example, it is said that the trend modelled with loess needs to be parametrized. No reference is given and it is unclear what is meant by this, as loess is a non-parametric regression methods and does not need a parametrization.

- This has been clarified L213.

Referencing within the article is not clear. Often Figures are referenced already in the methods description, making the reading difficult. A better separation of method and results would be helpful

- I am still not sure to understand this comment. There are no elements of the methods in the results, however figures are referenced in both sections. Please let me know the specific figures or lines where this problem appears.

Also sections called context and method are not clearly divided.

- Fixed.

**Reviewer #2**

    (a)   clarify and discuss the assumptions on the data series.

(a) The basic assumption is that the dataset is an additive signal of a long-term trend, a periodic anomaly (termed "cylce" in the manuscript) that does not change with time, observational noise and outliers. As demonstrated, the method is already useful for such cases. However, of to be even more useful, the authors should think about about extending the method to infer or take into account changes of the anomaly with time. At least they need to give the possibility to the user to supply a mask slicing the time series into chunks where the anomaly can be assumed constant, e.g., stacking winter/spring/summer into different stacks.

In the first of the three examples, the daily cycle of temperature (luckily) does not vary. But what about synoptic cycles? During clear-sky weeks the daily temperature cycle will be larger than on cloudy weeks. For signals influenced by vegetation, the cycles will differ with phenology, etc.

I tried applying the method to several soil respiration time series of the publicly available COSORE dataset. I got it to work technically, but was not able to properly detect outliers and aggregate to annual values. For some series, there was probably too few data within periods (Vern series: 4hourly measurement within a daily period), for others the properties of the signal changed too strongly with season (Migliavacca series).

The authors need to better clarify the assumption and limitations of the method. The method is not as general as claimed in the first version of the manuscript.

- The limits of **ctbi** have been acknowledged in the section "**Limits & recommendations**", L408-429. Following my former comment on this point:
  - (i) Residuals showing non-stationary variance (complex seasonality) can reasonably be well handled by **ctbi** if the original timeseries is segmented into bins of similar variability (quantified by the MAD). This has been performed on the MIGLIAVACCA dataset and added to the supplementary material.
  - (ii) Concerning cycles that stack on each other (daily cycle + weekly cycle + annual cycle + decadal cycle), the whole idea of **ctbi** was to use the aggregation as a tool to progressively remove high frequencies in the original signal. Successive aggregations will unfold these different periodicities.
  - (iii) Another limit is related to your next comment (b), which is the periods of instrument failure or human errors. **Ctbi** is not capable to make a distinction between a physically consistent signal and a random noise.

(b) The application of case-specific outlier-detection and aggregation is discussed as being a thing one wants to avoid. However, usually researchers know their data quite well and know their distributions, stability over time, problematic periods, changes in measurement equipment etc. It needs a more balanced discussion on the value of consistency for meta-analysis and usage of expert-knowledge.

- the importance of expert-knowledge has been acknowledged in the introduction (L24-27) and limits & recommendations (L408-410), as well as the conclusion (L444) in order to balance the discussion.

Outlook: many observational time series come in replicates. Can you think of ways to extend the method to use information across the replicates?

- It is difficult to design a generic script that can achieve that because most data are not standardized (different timestamps for the replica for example). I think it is much safer to let the user code the program to handle replicas by using **ctbi** as a tool. Exactly like I did for the complex seasonality problem.

Thank you!

François Ritter

---

## Referee Report (RR1)

Detection/removal of outliers is critical because – as the author correctly states – the raw data are altered thus impacting any subsequent statistical analysis. A widely accepted description of outliers as "an observation (or subset of observations) which APPEARS to be inconsistent with the remainder of that set of data" (Barnett&Lewis 1994) stresses the notorious difficulty to unambiguously identify a data point as an outlier and there is always the choice between correcting the data OR correcting the model. The problem is most eminent if there are few data – then there is a great chance that data are removed as outliers just because the assumed model is misspecified (and we must fear that this practice is wide-spread). The author considers the problem of outlier detection in the context of time series where the amount of data is typically relatively large. This opens the chance that we have a lot of information about the underlying distribution (the model) and we can exploit that knowledge to identify inconsistent observations. The approach chosen by the author addresses exactly this situation and the proposed modification of the original boxplot-idea seems to be very promising and robust. However, in the present form the implementation of the idea is not consistent. Roughly speaking, the author compares his approach with several older suggestions which are more or less inspired by the Gaussian case and he replaces these by a suggestion (k= 0.6) which is based on an average(median) of reasonable models. This will almost certainly lead to an improvement since the Gaussian case is an extreme case in the Skewness-Kurtosis plane. But in any particular application we don't have an average of distributions – we have one particular distribution and we should use an outlier detection method which is tailored to this particular distribution as good as possible. The author notes in his discussion of the precipitation use case: "… k=0.6 is insufficient here. A value of k ~ 5 is optimum in this case …". As the authors goal is to provide a "generic pre-processing procedure implemented in R" I would strongly advise to choose the value of k in a data-driven way instead of relying on a default value of k=0.6 thus introducing a quasi-standard even if there is the opportunity for the user to change this value. As far as I see, all the required knowledge is there. My idea would be to robustly estimate skewness and kurtosis from the data and then choose a value for k derived from the corresponding Pearson distribution. This would probably lead to a substantial improvement in robustness of the ctbi method and would also justify the publication of the LogBox approach in a more statistically oriented journal.

I am not a specialist in robust estimation but Kim&White (2004) could probably provide some ideas on how to estimate the kurtosis.

More detailed comments:

- The fact, that the Pearson family provides a distribution for any theoretically possible combination of skewness and kurtosis could be more explicitly stated. Also the fact that all the mentioned families (Beta,Gamma, …) are Pearson families and only later got their current names could be worth mentioning.
- The argument with the $3\sigma$, $4\sigma$ and $5\sigma$ convention seems relatively arbitrary. Only together with the requirement that the absolute number of erroneously flagged outliers should be restricted this makes sense. This important information is hidden only in the caption to Table 1. Perhaps you could argue the other way around starting from the expected number of erroneously flagged outliers. This would of course change the regression equations (If you go for exactly one erroneously flagged outlier in the Gaussian case the equation should roughly be -0.5 + 0.322 log(n) )
- I don't see the need to present the results for the generalized Gamma distribution as the bounded support precludes its use a priori.
- You are looking for methods which are adapted to particular distributions. A general procedure to deal with non-normal data, which is also popular in the time-series context, is

the transformation (log, square root, Box-Cox, …) of the data before statistical analysis. It would be worth comparing the transformation approach with the LogBox approach when detecting outliers.

- I personally find ctbi a bit cryptic. How about CTBin which at least makes the main ingredient – the bins – visible.

Barnett, V. and T. Lewis (1994) Outliers in Statistical Data. 3rd ed. Wiley&Sons

Kim, T.-H. and H. White (2004) On more robust estimation of skewness and kurtosis. Finance Research Letters 1 (1)

---

## Referee Report (RR2)

I see considerable improvement in the manuscript. The main advantages in the current version of the Logbox method are in my opinion:

1. The relationship between the choice of $\alpha(n)$ and the expected number of erroneously flagged outliers is now made obvious with the introduction of the function $f(n)$.
2. The procedure is now adaptive to the type of underlying distribution by introducing the functions $g_A$ and $g_B$, depending on a sample estimate of $m_*$.

The inclusion of heavy-tailed extreme value distributions is reasonable to extend the range of applicability of the Logbox method, although in the case of very heavy-tailed distributions probably transformations would be applied prior to outlier detection.

Due to the introduction of the two pragmatically chosen and fitted nonlinear functions $g_A$ and $g_B$ the resulting procedure is not very elegant but I accept the authors argument that the established relationship with the well-known boxplot procedure probably contributes to the acceptance of the method.

I strongly suggest that the newly introduced part concerning outlier detection for very small samples ($n < 9$) will be removed from the paper for at least two reasons:

1. In the envisaged application of the outlier detection method as a preprocessing tool in time series analysis such small sample sizes are irrelevant.
2. As I stressed already in my first review, outlier detection is even more critical with small sample sizes – and I am deeply convinced that in samples as small as $n = 8$ one should never use statistical arguments to declare single observations as outliers.

---

## Author Response (AR2)

Dear Anke Hildebrandt and dear Jens Schumacher,

I managed to update the Logbox method so that the outlier cutting threshold is adapted to the shape and size of the data. For the past two years, I have thought that this was not feasible due to a large risk of overfitting at small/moderate samples sizes. However, the results are surprisingly good (I used data from 6307 century-old weather stations to gain confidence in the method) and I have to thank Dr. Schumacher for encouraging me in this attempt. Part I has been entirely reshaped, and part II has been positively impacted by these changes. Table 1 disappeared, all figures have been updated, the supplementary material as well. The manuscript is clearer and better.

I explain the major steps that led to this update in the following:

**1) The family of distributions required to parametrize the model**. The Pearson family represents light-tailed samples (Beta has been excluded due to a bounded support). For heavy-tailed samples (ignored in the original manuscript), I chose the Generalized Extreme Value family (Weibull, Fréchet & Gumbel) usually used to model the behavior of extrema. I have access to the quantile function Q of each distribution with high accuracy.

**2) The framework.** In the original manuscript, I picked $\pm 3/4/5\sigma$ cutting thresholds and associated them a sample size of $10^2/10^4/10^6$ to fit a line and find $\alpha = k\log(n) + 1$ (the intercept was forced to be 1 to simplify the model). As Jens Schumacher suggested, I replaced this discrete framework with $\alpha(n) = \frac{Q\left(1 - \frac{f(n)}{2n}\right) - Q(0.75)}{Q(0.75) - Q(0.25)}$, with $f$ a continuous function that gives the number of erroneously flagged outliers: $f(n) = 0.001\sqrt{n}$. For example, for a sample size of $10^6$ I am expecting to cut $f(10^6) = 1$ point. For each distribution, I computed $\alpha(n)$ versus $n$ for five samples ($n = 10^2, 10^3, 10^4, 10^5, 10^6$) and found that $\alpha(n) = A\log(n) + B$ is an accurate model for both the Pearson & GEV family ($r^2 = 0.994 \pm 0.005$ & $r^2 = 0.99 \pm 0.01$). Now there are two known parameters (A,B) for each distribution, and you can notice that $f(n)$ is **not** a flat number, otherwise alpha would become arbitrary large.

**3) Case of small sample sizes.** The original manuscript ignored biases emerging in small samples. This has been fixed by adding $\frac{C}{n}$ to $\alpha$ in a similar manner than Carling et al. (2000): $\alpha(n) = A\log(n) + B + \frac{C}{n}$. The constant value of $C = 36$ had been determined with small random samples ($n = 9$) to limit type I errors to 0.1%.

**4) Model to determine A and B on an unknown sample**. The centered Moors $m = m_- + m_+ - 1.23$ is a predictor of the kurtosis excess (Moors 1988, Kim & White 2004) using $m_- = (E_3 - E_1)/(E_6 - E_2)$ and $m_+ = (E_7 - E_5)/(E_6 - E_2)$ with $E_i = q(i/8)$ the sample octile. This study introduces a modified version defined as $m_* = \max(m_-, m_+) - 0.6165$ because $m_*$ is more appropriate than $m$ to determine if a sample is light-tailed or heavy-tailed. For example, a Gaussian distribution ($m_- = m_+ \approx 0.6165$ ; $m = m_* \approx 0$) and a right-skewed distribution with one heavy tail ($m_- = 0.1$ and $m_+ = 1.13$) will share identical $m$ but different $m_*$. This difference between $m$ and $m_*$ explains why my attempts to construct a model adapted to the shape of the residuals failed in the past. Figure 1e,f shows the relationship $A$ versus $m_*$ and $B$ versus $m_*$ for all distributions, with their best fit, $g_A(m_*) = 0.2294e^{2.9416m_* - 0.0512m_*^2 - 0.0684m_*^3}$ ($r^2 = 0.999$) and $g_B(m_*) = 1.0585 + 15.6960m_* - 17.3618m_*^2 + 28.3511m_*^3 - 11.4726m_*^4$ ($r^2 = 0.999$).

**5) Logbox model.** The Logbox model is finally $\alpha(n) = g_A(m_*)\log(n) + g_B(m_*) + \frac{36}{n}$ for $n \geq 9$. For $3 \leq n \leq 8$, the cutting thresholds are computed using the MAD (safer breakdown point, see method).

**6) Testing Logbox and comparing it with other methods in the literature.** In the original manuscript, the model was tested on the same theoretical distributions used to parametrize it, without connection with real residuals obtained in Earth Science (this was a critic of the reviewer #1). I changed this and downloaded all the data available for the oldest weather stations on Earth (6307 stations with more than 100 years of daily precipitation and temperature). The residuals have been extracted and the suspicious values flagged by the NOAA have been discarded. The observed percentage of flagged outliers is shown in Fig. 2, and I can compare it with the theoretical percentage: $p_{theo} = f(n) \times \frac{100}{n} = \frac{0.1}{\sqrt{n}}\%$. Results went beyond my most optimistic expectation for both small and moderate samples.

**7) Changes in part II.** There only is one change in the outlier level used in the precipitation. Now that the behavior of heavy-tailed distributions has been explored in Part I, it appears that the outlier level formerly

chosen for the precipitation dataset (30 years of daily data) was too *low*: $y_{outlier} = y_{max} + \frac{1}{2}(y_{max} - \mu)$ with $y_{max}$ and $\mu$ respectively the maximum and mean of the dataset. These $y_{outlier}$ values are in fact statistically plausible, and coincidently correspond to the cutting threshold computed by the new Logbox procedure (it has been proved in part I that the cutting threshold was correctly produced by Logbox). In order to choose a less arbitrary outlier level, I applied the following procedure: For each station with daily precipitation over 100 years, I considered that a precipitation event 20% above the century maximum is "impossible" : $y_{outlier} = 1.2 \times (y_{max})_{100\ years}$. Then I randomly selected 30 years within each station $i$ to compute $\lambda_i = \frac{y_{outlier}}{(y_{max})_{30\ years}}$. The mean value for all stations is $\lambda = 1.6 \pm 0.4$, leading to $y_{outlier} = 1.6 \times y_{max}$ that is less arbitrary than $y_{outlier} = y_{max} + \frac{1}{2}(y_{max} - \mu)$. This change only affects 1 false negative for the Logbox procedure, but does not affect the *tsoutlier* function which fails at capturing outliers anyway.

**Point-by-point answer**

Detection/removal of outliers is critical because – as the author correctly states – the raw data are altered thus impacting any subsequent statistical analysis. A widely accepted description of outliers as "an observation (or subset of observations) which APPEARS to be inconsistent with the remainder of that set of data" (Barnett&Lewis 1994) stresses the notorious difficulty to unambiguously identify a data point as an outlier and there is always the choice between correcting the data OR correcting the model. The problem is most eminent if there are few data – then there is a great chance that data are removed as outliers just because the assumed model is misspecified (and we must fear that this practice is wide-spread). The author considers the problem of outlier detection in the context of time series where the amount of data is typically relatively large. This opens the chance that we have a lot of information about the underlying distribution (the model) and we can exploit that knowledge to identify inconsistent observations. The approach chosen by the author addresses exactly this situation and the proposed modification of the original boxplot-idea seems to be very promising and robust. However, in the present form the implementation of the idea is not consistent. Roughly speaking, the author compares his approach with several older suggestions which are more or less inspired by the Gaussian case and he replaces these by a suggestion (k= 0.6) which is based on an average(median) of reasonable models. This will almost certainly lead to an improvement since the Gaussian case is an extreme case in the Skewness-Kurtosis plane. But in any particular application we don't have an average of distributions – we have one particular distribution and we should use an outlier detection method which is tailored to this particular distribution as good as possible. The author notes in his discussion of the precipitation use case: "… k=0.6 is insufficient here. A value of k ~ 5 is optimum in this case …". As the authors goal is to provide a "generic pre-processing procedure implemented in R" I would strongly advise to choose the value of k in a data-driven way instead of relying on a default value of k=0.6 thus introducing a quasi-standard even if there is the opportunity for the user to change this value. As far as I see, all the required knowledge is there. My idea would be to robustly estimate skewness and kurtosis from the data and then choose a value for k derived from the corresponding Pearson distribution. This would probably lead to a substantial improvement in robustness of the ctbi method and would also justify the publication of the LogBox approach in a more statistically oriented journal. I am not a specialist in robust estimation but Kim&White (2004) could probably provide some ideas on how to estimate the kurtosis.

[…]

● The argument with the 3σ, 4σ and 5σ convention seems relatively arbitrary. Only together with the requirement that the absolute number of erroneously flagged outliers should be restricted this makes sense. This important information is hidden only in the caption to Table 1. Perhaps you could argue the other way around starting from the expected number of erroneously flagged outliers. This would of course change the regression equations (If you go for exactly one erroneously flagged outlier in the Gaussian case the equation should roughly be -0.5 + 0.322 log(n) )

I have entirely updated the manuscript to follow these suggestions (see above).

● The fact, that the Pearson family provides a distribution for any theoretically possible combination of skewness and kurtosis could be more explicitly stated. Also the fact that all the mentioned families (Beta,Gamma, …) are Pearson families and only later got their current names could be worth mentioning.

This has been updated in the introduction of Part I.

● I don't see the need to present the results for the generalized Gamma distribution as the bounded support precludes its use a priori.

The Generalized Gamma Distribution has been removed from the supplementary material and discussion. I also removed the Beta distribution from the Pearson family due to the same problems emerging with the bounded support.

● You are looking for methods which are adapted to particular distributions. A general procedure to deal with non-normal data, which is also popular in the time-series context, is the transformation (log, square root, Box-Cox, …) of the data before statistical analysis. It would be worth comparing the transformation approach with the LogBox approach when detecting outliers.

Data transformations such as the Box-Cox method have been used for comparison in part II.

• I personally find ctbi a bit cryptic. How about CTBin which at least makes the main ingredient – the bins – visible.

I unfortunately need to stick to the ctbi name as it has already been created on the CRAN.

**Important details**

- All the code has been updated in https://github.com/fritte2/ctbi_article
- The 6307 stations can be downloaded and the residuals can be extracted with this code, however this will take ~40 hours of computing. Instead, I can share the data with Jens Schumacher on a google drive link (~8 Gb).
- The new ctbi version has not been updated on the CRAN yet, but all the changes are available on https://github.com/fritte2/ctbi. I will upload the new ctbi version once I receive feedbacks on the manuscript.
- My affiliation has changed (for the last time!) : « Laboratoire des Sciences du Climat et de l'Environnement, LSCE/IPSL, CEA-CNRS-UVSQ, Université Paris-Saclay, Gif-sur-Yvette 91191, France »
- The high quality figures are available in .PDF. The figures shown in the manuscript are only poor quality snapshots (Microsoft Word has problem to incorporate PDF). Will the editorial staff be able to include the original PDF figures in the article? Thank you!

**References**

Carling, Kenneth. "Resistant outlier rules and the non-Gaussian case." Computational Statistics & Data Analysis 33, no. 3 (2000): 249-258.

Kim, Tae-Hwan, and Halbert White. "On more robust estimation of skewness and kurtosis." *Finance Research Letters* 1, no. 1 (2004): 56-73.

Moors, J. J. A. "A quantile alternative for kurtosis." *Journal of the Royal Statistical Society: Series D (The Statistician)* 37, no. 1 (1988): 25-32.

---

## Author Response (AR3)

Dear Anke Hildebrandt and dear Jens Schumacher,

Thank you for all the time you put in this manuscript. I have adressed the changes asked by the reviewer and removed the very small sample case (n < 9) from the Logbox procedure. Another change came from an independant reviewer (Dr. Rob Hyndman), who mentioned that an important reference was missing: Barbato et al. (2011).

I was not aware of the study from Barbato et al. (2011), but it appears that they found a law similar to logbox (alpha = A*log(n)+B) but with a different reasoning (heuristic approach) and calibrated on the Gaussian distribution only (A and B are constant). I had to include this approach in part I, and to compare its performance to Logbox (Fig. 2 updated). The discussion has been slightly modified to include this new model, but the conclusion remains the same.

Please find below a list of changes:

**Part I:**

- the very small sample case (n < 9) has been removed and the method section has therefore been simplified.

- m.star is not exactly a predictor of the kurtosis excess (which takes into account the two tails of a distribution), but it is more a predictor of the weight of the heavier tail. The description of m.star has been updated accordingly.

- Barbato et al. (2011) has been included in the introduction and the discussion.

**Part II:** unaffected

**References:** updated to the HESS format.

**supplementary material:**

- The very small sample case (n < 9) has been removed, and Fig. S1 updated.

**Acknowledgement:**

- I will personally fund the publication of this article, and I therefore removed the "fonds de dotation O" from the acknowledgement.

**Code**

- the code has been updated on https://github.com/fritte2/ctbi_article to account for the changes in Part I.

Ref:

Barbato, G., Barini, E. M., Genta, G., and Levi, R.: Features and Performance of Some Outlier Detection Methods, Journal of Applied Statistics, https://doi.org/10.1080/02664763.2010.545119, 2011.

Best regards,

François Ritter